# A SECOND-ORDER-LIKE OPTIMIZER WITH ADAPTIVE GRADIENT SCALING FOR DEEP LEARNING

## ABSTRACT

In this empirical article, we introduce INNAprop, an optimization algorithm that combines the INNA method with the RMSprop adaptive gradient scaling. It leverages second-order information and rescaling while keeping the memory requirements of standard DL methods as AdamW or SGD with momentum. After giving geometrical insights, we evaluate INNAprop on CIFAR-10, Food101, and ImageNet with ResNets, VGG, DenseNet, and ViT, and on GPT-2 (OpenWeb-Text) train from scratch and with LoRA fine-tuning (E2E). INNAprop consistently matches or outperforms AdamW both in training speed and accuracy, with minimal hyperparameter tuning in large-scale settings. Our code is publicly available at https://github.com/innaprop/innaprop.

## 1 INTRODUCTION

As deep learning models grow in size, massive computational resources are needed for training, representing significant challenges in terms of financial costs, energy consumption, and processing time (Susnjak et al., 2024; Varoquaux et al., 2024). According to the UN's Environment Programme Training, the Big Tech sector produced between two and three percent of the world's carbon emissions in 2021; some estimations for the year 2023 go beyond 4%, see the latest Stand.earth reports, and also (Schwartz et al., 2020; Strubell et al., 2020; Patterson et al., 2021) for related issues. For instance, training GPT-3 is estimated to require 1,287 megawatt-hours (MWh) of electricity, equivalent to the annual usage of over 100 U.S. households (Anthony et al., 2020; Patterson et al., 2021). Similarly, the financial cost of specialized hardware and cloud computing is extremely high. OpenAI claimed that the training cost for GPT-4 (Achiam et al., 2023) exceeded 100 million dollars. The PaLM model developed by Google AI was trained for two months using 6144 TPUs for 10 million dollars (Chowdhery et al., 2023). All this implies a need for faster and more cost-efficient optimization algorithms. It also suggests that early stopping (Prechelt, 2002; Bai et al., 2021) in the training phase is a desirable feature whenever possible.

We focus in this work on computational efficiency during the training phase and consider the problem of unconstrained minimization of a loss function $\mathcal{J}: \mathbb{R}^p \to \mathbb{R}$, as follows

$$\min_{\theta \in \mathbb{R}^p} \mathcal{J}(\theta). \tag{1}$$

**Continuous dynamical systems as optimization models.** To achieve higher efficiency, it is necessary to deeply understand how algorithms work and how they relate to each other. A useful way to do this is by interpreting optimization algorithms as discrete versions of continuous dynamical systems (Ljung, 1977), further developed in (Harold et al., 1997; Benaïm, 2006; Borkar & Borkar, 2008; Attouch et al., 2016; Aujol et al., 2019). In deep learning, this approach is also quite fruitful; it has, in particular, been used to provide convergence proofs or further geometric insights (Davis et al., 2020; Bolte & Pauwels, 2020; Barakat & Bianchi, 2021; Chen et al., 2023a).

In the spirit of Castera et al. (2021), we consider the following continuous-time dynamical system introduced in Alvarez et al. (2002) and referred to as DIN (standing for "dynamical inertial Newton"):

$$\underbrace{\ddot{\theta}(t)}_{\text{Inertial term}} + \underbrace{\alpha\,\dot{\theta}(t)}_{\text{Friction term}} + \underbrace{\beta\,\nabla^2\mathcal{J}(\theta(t))\dot{\theta}(t)}_{\text{Newtonian effects}} + \underbrace{\nabla\mathcal{J}(\theta(t))}_{\text{Gravity effect}} = 0, \qquad t \geq 0, \tag{2}$$

where $t$ is the time, $\mathcal{J}\colon \mathbb{R}^p \to \mathbb{R}$ is a loss function to be minimized (e.g., empirical loss in DL applications) as in Equation (1), assumed $C^2$ with gradient $\nabla\mathcal{J}$ and Hessian $\nabla^2\mathcal{J}$. A key aspect of Equation (2) that places it between first- and second-order optimization is that a change of variables allows to describe it using only the gradient $\nabla\mathcal{J}$, since $\nabla^2\mathcal{J}(\theta(t))\dot\theta(t) = \frac{d}{dt}\nabla\mathcal{J}(\theta(t))$ (see Section 2.2 for details). This greatly reduces computational costs, as it can be discretized as a difference of gradients which does not require Hessian vector product, making it possible to design more practical algorithms, as shown in Chen & Luo (2019); Castera et al. (2021); Attouch et al. (2022).

We recover the continuous-time heavy ball system by assuming $\alpha > 0$, and removing the geometrical "damping" term in Equation (2) through the choice $\beta = 0$. A discrete version of this system corresponds to the Heavy Ball method (Polyak, 1964), which is at the basis of SGD solvers with momentum in deep learning (Qian, 1999; Sutskever et al., 2013). By allowing both $\alpha$ and $\beta$ to vary, we recover Nesterov acceleration (Nesterov, 1983; Su et al., 2016; Attouch et al., 2019).

**Adaptive methods.** Adaptive optimization methods, such as RMSprop (Tieleman et al., 2012) and AdaGrad (Duchi et al., 2011), modify the update dynamics by introducing coordinate-wise scaling of the gradient based on past information. These methods can be modeled by continuous-time ODEs of the following form, expressed here for the simple gradient system:

$$\dot\theta(t) + \frac{1}{\sqrt{G(t, \theta(t)) + \epsilon}} \odot \nabla\mathcal{J}(\theta(t)) = 0, \quad t \geq 0, \tag{3}$$

where $\epsilon > 0$, $G(t, \theta(t)) \in \mathbb{R}^p$ represents accumulated information. The scalar addition, square root, and division are understood coordinatewise and $\odot$ denotes the coordinate-wise product for vectors in $\mathbb{R}^p$. In AdaGrad or RMSprop, $G(t, \theta(t))$ is a gradient amplitude averaged of the form:

$$G(t, \theta(t)) := \int_0^t \nabla\mathcal{J}(\theta(\tau))^2 \, d\mu_t(\tau), \tag{4}$$

for different choices of $\mu_t$ — uniform for AdaGrad and moving average for RMSprop. This generally improves performance, see the pioneering work (Duchi et al., 2011; Tieleman et al., 2012).

**Our approach.** We combine the "dynamical inertial Newton" method (DIN) from Equation (2) with an RMSprop adaptive gradient scaling. This allows us to take into account second-order information for the RMSProp scaling. Computationally, this second-order information is expressed using a time derivative. In discrete time, this will provide a second-order intelligence with the same computational cost as gradient evaluation. The resulting continuous time ODE is given as follows:

$$\ddot\theta(t) + \alpha\,\dot\theta(t) + \beta\,\frac{d}{dt}\mathrm{RMSprop}(\mathcal{J}(\theta(t))) + \mathrm{RMSprop}(\mathcal{J}(\theta(t))) = 0, \qquad t \geq 0 \tag{5}$$

$$\text{where } \mathrm{RMSprop}(\mathcal{J}(\theta(t))) = \frac{1}{\sqrt{G(t, \theta(t)) + \epsilon}} \odot \nabla\mathcal{J}(\theta(t))$$

with $G$ of the form (4) with an adequate time-weight distribution $\mu_t$ corresponding to the RMSProp scaling. A discretization of this continuous time system, combined with careful memory management, results in our new optimizer INNAprop, see Section 2.1.

**Relation with existing work.** To improve the efficiency of stochastic gradient descent (SGD), two primary strategies are used: leverage local geometry for having clever directions and incorporate momentum to accelerate convergence. These approaches include accelerated methods (e.g., Nesterov's acceleration (Nesterov, 1983; Dozat, 2016), momentum SGD (Polyak, 1964; Qian, 1999; Sutskever et al., 2013), and adaptive methods (e.g., Adagrad (Duchi et al., 2011), RMSProp (Tieleman et al., 2012)), which adjust learning rates per parameter.

Adam remains the dominant optimizer in deep learning. It comes under numerous variants proposed to improve its performance or to adapt it to specific cases (Dozat, 2016; Shazeer & Stern, 2018; Reddi et al., 2019; Loshchilov & Hutter, 2017; Zhuang et al., 2020). Adafactor (Shazeer & Stern, 2018) improves memory efficiency, Lamb (You et al., 2019) adds layerwise normalization, and Lion (Chen et al., 2023b) uses sign-based momentum updates. AdEMAMix (Pagliardini et al., 2024) combines two EMAs, while Defazio et al. (Defazio et al., 2024) introduced a schedule-free method incorporating Polyak-Ruppert averaging with momentum.

One of the motivations of our work is the introduction of second-order properties in the dynamics akin to Newton's method. Second-order optimizers are computationally expensive due to frequent Hessian computations (Gupta et al., 2018; Martens & Grosse, 2015). Their adaptation to large scale learning settings require specific developments (Jahani et al., 2021; Qian et al., 2021). For example, the Sophia optimizer (Liu et al., 2023), designed for large language models, uses a Hessian-based pre-conditioner to penalize high-curvature directions. In this work, we draw inspiration from INNA (Castera et al., 2021), based on the continuous time dynamics introduced by (Alvarez et al., 2002), which combines gradient descent with a Newtonian mechanism for first-order stochastic approximations.

Our proposed method, INNAProp, integrates the algorithm INNA, which features a Newtonian effect with cheap computational cost, with the gradient scaling mechanism of RMSprop. This preserves the efficiency of second-order methods and the adaptive features of RMSprop while significantly reducing the computational overhead caused by Hessian evaluation. Specific hyperparameter choices for our method allow us to recover several existing optimizers as special cases.

**Contributions.** They can be summarized as follows:

- We introduce INNAprop, a new optimization algorithm that combines the Dynamical Inertial Newton (DIN) method with RMSprop's adaptive gradient scaling, efficiently using second-order information for large-scale machine learning tasks. We obtain a second-order optimizer with computational requirements similar to first-order methods like AdamW, making it suitable for deep learning (see Section 2.2 and Appendix B).

- We provide a continuous-time explanation of INNAprop, connecting it to second-order ordinary differential equations (see Section 2 and Equation (5)). We discuss many natural possible discretizations and show that INNAprop is empirically the most efficient. Let us highlight a key feature of our method: it incorporates second-order terms in space without relying on Hessian computations or inversions of linear systems which are both prohibitive in deep learning.

- We show through extensive experiments that INNAprop matches or outperforms AdamW in both training speed and final accuracy on benchmarks such as image classification (CIFAR-10, ImageNet) and language modeling (GPT-2) (see Section 3).

We describe our algorithm and its derivation in Section 2. Hyperparameter tuning recommendations and our experimental results are provided in Section 3.

## 2 INNAprop: a second-order method in space and time based on RMSprop

### 2.1 The algorithm

Our method is built on the following Algorithm 1, itself derived from a combination of INNA (Castera et al., 2021) and RMSprop (Tieleman et al., 2012) (refer to Section 2.2 for more details). The following version of the method is the one we used in all experiments. It includes the usual ingredients of deep-learning training: mini-batching, decoupled weight-decay, and scheduler procedure. For a simpler, "non-deep learning" version, refer to Algorithm 2 in Appendix B.

In Algorithm 1, SetLrSchedule is the "scheduler" for step-sizes; it is defined as a custom procedure for handling learning rate sequences for different networks and databases. To provide a full description of our algorithm, we provide detailed explanations of the scheduler procedures used in our experiments (Section 3) in Appendix D, along with the corresponding benchmarks.

**Remark 1 (Well posedness)** Observe that, for all schedulers $\gamma_k < \beta$ for $k \in \mathbb{N}$, so that INNAprop is well-posed (line 13 in Algorithm 1, the division is well defined).

### 2.2 Derivation of the algorithm

There are several ways to combine RMSprop and INNA, or DIN its second-order form, as there exist several ways to do so with the heavy ball method and RMSprop. We opted for the approach below because of its mechanical and geometrical appeal and its numerical success (see Appendix B for

---

**Algorithm 1** Deep learning implementation of INNAprop

---

1: **Objective function:** $\mathcal{J}(\theta) = \frac{1}{n} \sum_{n=1}^{N} \mathcal{J}_n(\theta)$ for $\theta \in \mathbb{R}^p$.
2: **Learning step-sizes:** $\gamma_k := \{\text{SetLrSchedule}(k)\}_{k \in \mathbb{N}}$ where $\gamma_0$ is the initial learning rate.
3: **Hyper-parameters:** $\sigma \in [0, 1]$, $\alpha \geq 0$, $\beta > \sup_{k \in \mathbb{N}} \gamma_k$, $\lambda \geq 0$, $\epsilon = 10^{-8}$.
4: **Mini-batches:** $(\mathsf{B}_k)_{k \in \mathbb{N}}$ of nonempty subsets of $\{1, \ldots, N\}$.
5: **Initialization:** time step $k \leftarrow 0$, parameter vector $\theta_0$, $v_0 = 0$, $\psi_0 = (1 - \alpha\beta)\theta_0$.
6: **for** $k = 1$ **to** K **do**
7:     $\boldsymbol{g}_k = \frac{1}{|\mathsf{B}_k|} \sum_{n \in \mathsf{B}_k} \nabla \mathcal{J}_n(\boldsymbol{\theta}_k)$          $\triangleright$ select batch $\mathsf{B}_k$ and return the corresponding gradient
8:     $\gamma_k \leftarrow \text{SetLrSchedule}(k)$                                         $\triangleright$ see above and Remark 1
9:     $\boldsymbol{\theta}_k \leftarrow (1 - \lambda\gamma_k)\boldsymbol{\theta}_k$                                                   $\triangleright$ decoupled weight decay
10:    $\boldsymbol{v}_{k+1} \leftarrow \sigma\boldsymbol{v}_k + (1 - \sigma)\boldsymbol{g}_k^2$
11:    $\hat{\boldsymbol{v}}_{k+1} \leftarrow \boldsymbol{v}_{k+1}/(1 - \sigma^k)$
12:    $\boldsymbol{\psi}_{k+1} \leftarrow \left(1 - \frac{\gamma_k}{\beta}\right)\boldsymbol{\psi}_k + \gamma_k \left(\frac{1}{\beta} - \alpha\right)\boldsymbol{\theta}_k$
13:    $\boldsymbol{\theta}_{k+1} \leftarrow \left(1 + \frac{\gamma_k(1 - \alpha\beta)}{\beta - \gamma_k}\right)\boldsymbol{\theta}_k - \frac{\gamma_k}{\beta - \gamma_k}\boldsymbol{\psi}_{k+1} - \gamma_k\beta \left(\boldsymbol{g}_k/(\sqrt{\hat{\boldsymbol{v}}_{k+1}} + \epsilon)\right)$
14: **return** $\boldsymbol{\theta}_{K+1}$

---

further details). Consider the following dynamical inertial Newton method (Alvarez et al., 2002):

$$\ddot{\theta}(t) + \alpha \dot{\theta}(t) + \beta \frac{d}{dt}\nabla\mathcal{J}(\theta(t)) + \nabla\mathcal{J}(\theta(t)) = 0, \quad t \geq 0, \tag{6}$$

as in Equation (2) and replacing $\nabla^2\mathcal{J}(\theta(t))\dot{\theta}(t)$ by $\frac{d}{dt}\nabla\mathcal{J}(\theta(t))$. Using finite differences with a fixed time step $\gamma$ for discretization, replacing in particular the gradient derivatives by gradient differences:

$$\frac{d}{dt}\nabla\mathcal{J}(\theta(t)) \simeq \frac{\nabla\mathcal{J}(\theta_{k+1}) - \nabla\mathcal{J}(\theta_k)}{\gamma},$$

where $\theta_k, \theta_{k+1}$ correspond to two successive states around the time $t$.

Setting $\nabla\mathcal{J}(\theta_k) = g_k$, we obtain $\dfrac{\theta_{k+1} - 2\theta_k + \theta_{k-1}}{\gamma} + \alpha\dfrac{\theta_k - \theta_{k-1}}{\gamma} + \beta\dfrac{g_k - g_{k-1}}{\gamma} + g_{k-1} = 0.$

To provide our algorithm with an extra second-order geometrical intelligence, we use the proxy of RMSprop direction in place of the gradient.

Choose $\sigma > 0$ and $\epsilon > 0$, and consider:

$$v_{k+1} = \sigma v_k + (1 - \sigma)g_k^2 \tag{7}$$

$$\frac{\theta_{k+1} - 2\theta_k + \theta_{k-1}}{\gamma} + \alpha\frac{\theta_k - \theta_{k-1}}{\gamma} + \beta\frac{\frac{g_k}{\sqrt{v_{k+1}}+\epsilon} - \frac{g_{k-1}}{\sqrt{v_k}+\epsilon}}{\gamma} + \frac{g_{k-1}}{\sqrt{v_k} + \epsilon} = 0. \tag{8}$$

Although this system has a natural mechanical interpretation, its memory footprint is abnormally important for this type of algorithm: for one iteration of the system (7)-(8), it culminates at 6 full dimension memory slots, namely $g_{k-1}, g_k, \theta_{k-1}, \theta_k, v_k$, and $v_{k+1}$ before the evaluation of (8).

Therefore, we proceed to rewrite the algorithm in another system of coordinates. The computations and the variable changes are provided in Appendix B. We eventually obtain:

$$v_{k+1} = \sigma v_k + (1 - \sigma)g_k^2$$

$$\psi_{k+1} = \psi_k \left(1 - \frac{\gamma}{\beta}\right) + \gamma \left(\frac{1}{\beta} - \alpha\right)\theta_k,$$

$$\theta_{k+1} = \left(1 + \frac{\gamma(1 - \beta\alpha)}{\beta - \gamma}\right)\theta_k - \frac{\gamma}{\beta - \gamma}\psi_{k+1} - \gamma\beta\frac{g_k}{\sqrt{v_{k+1}} + \epsilon}$$

which only freezes 3 full dimension memory slots corresponding to $v_k, \psi_k, \theta_k$. As a result, the memory footprint is equivalent to that of the Adam optimizer (see Table 5).

**Remark 2 (On other possible discretizations)** (a) If we use the proxy of RMSprop directly with INNA (Castera et al., 2021), we recover indeed INNAprop through a rather direct derivation (see

Appendix C.1 for more details). Our motivation to start from the "mechanical" version of the algorithm is to enhance our understanding of the geometrical features of the algorithm.

(b) RMSprop with momentum (Graves, 2013) is obtained by a discretization of the heavy ball continuous time system, using a momentum term and an RMSprop proxy. It would be natural to proceed that way in our case, and it indeed leads to a different method (see Appendix C.2). However, the resulting algorithm appears to be numerically unstable (see Figure 10 for an illustration).

(c) Incorporating RMSprop as it is done in Adam using momentum leads to a third method (see Appendix C.3), which appears to be extremely similar to NAdam Dozat (2016); it was thus discarded.

**Remark 3 (A family of algorithms indexed by $\alpha, \beta$)** INNAprop can be seen as a family of methods indexed by the hyperparameters $\alpha$ and $\beta$. When $\beta = 0$, we recover a modified version of RMSprop with momentum (Graves, 2013) (see Appendix B.1). For $\alpha = \beta = 1$, INNAprop with its default initialization, boils down to AdamW without momentum ($\beta_1 = 0$), see Appendix B.1 and Table 5. By setting $\alpha = \beta = 1$, we empirically recover the behavior of AdamW. Experiments demonstrate that this consistently aligns with AdamW, suggesting that AdamW can be seen as a special case within the broader INNAprop family. See Appendix B.1 for further details and illustrations. We now explain how these hyperparameters $(\alpha, \beta)$ have been tuned on "small size" problems.

## 3 EMPIRICAL EVALUATION OF INNAPROP

We conduct extensive comparisons of the proposed algorithm and the AdamW optimizer, which is dominantly used in image classification (Chen et al., 2018; Zhuang et al., 2020; Touvron et al., 2021; Mishchenko & Defazio, 2023) and language modeling tasks (Brown et al., 2020; Hu et al., 2021; Liu et al., 2023). Hyperparameter tuning (Sivaprasad et al., 2020) is a crucial issue for this comparison, and we start with this. As a general rule, we strive to choose the hyperparameters that give a strong baseline for AdamW (based on literature or using grid search). Unless stated differently, our experiments use the AdamW optimizer [1] with its default settings as defined in widely-used libraries (Paszke et al., 2019; Bradbury et al., 2018; Abadi et al., 2016): $\beta_1 = 0.9$, $\beta_2 = 0.999$, $\lambda = 0.01$ and $\epsilon = 1e - 8$. For INNAprop, unless otherwise specified, the default settings for the RMSprop component align with those of AdamW: $\sigma = 0.999$ and $\epsilon = 1e - 8$.

The INNAprop method and the AdamW optimizer involve different classes of hyperparameters; some of them are common to both algorithms, and some are specific.

*Our hyperparameter tuning strategy for both algorithms is summarized in Table 1.*

We begin this section with the tuning of parameters $\alpha, \beta$ for INNAprop on CIFAR10 with VGG and ResNet architectures and then use these parameters on larger datasets and models. We use as much as possible the step size scheduler and weight decay settings reported in the literature for the AdamW optimizer, which we believe to be well-adjusted and provide adequate references for each experiment. These are used both for AdamW and INNAprop. With this protocol, we only perform minimal hyperparameter tuning for INNAprop for larger-scale experiments. This is due to constrained computational resources. We aim to demonstrate the typical performance of the Algorithm 1, rather than its peak performance with extensive tuning.

Table 1: Hyperparameter tuning strategy for INNAprop and AdamW: AdamW is systematically favored.

| Parameters | AdamW tuning | INNAprop tuning | Comparative advantage |
|---|---|---|---|
| Learning rate | Literature or grid search tuning | Reused from AdamW | AdamW favored |
| Step size scheduler | Literature | Reused from AdamW | N/A |
| Weight decay | Literature or grid search tuning | Reused from AdamW | AdamW favored |
| RMSprop parameter | Default or literature | Reused from AdamW | AdamW favored |
| Inertial parameters $(\alpha, \beta)$ | N/A | Tuned on CIFAR-10 | N/A |

---

[1]`https://pytorch.org/docs/stable/generated/torch.optim.AdamW.html`

### 3.1 TUNING INNAPROP ON CIFAR-10 WITH VGG11 AND RESNET18

**Hyperparameter tuning:** We tune $(\alpha, \beta)$ using VGG11 (Simonyan & Zisserman, 2014) and ResNet18 (He et al., 2016) models trained on CIFAR10 (Krizhevsky & Hinton, 2010), together with the initial learning rate $\gamma_0$ to ensure proper training. We fix a cosine scheduler where $T_{\max} = 200$ and $\gamma_{\min} = 0$ (see Appendix D for more details) and we consider two weight decay parameters $\lambda = 0$ or $\lambda = 0.01$. Our experiment suggests using an initial learning rate $\gamma_0 = 10^{-3}$, which is the baseline value reported for AdamW in this experiment (see Appendix E). For INNAprop, we optimize only the hyperparameters $\alpha$ and $\beta$, using test accuracy and training loss as the optimization criteria. A grid search is performed over $(\alpha, \beta) \in \{0.1, 0.5, 0.9, \dots, 3.5, 4.0\}$ using optuna (Akiba et al., 2019). In Figure 1, we detail the obtained training loss and test accuracy for various $(\alpha, \beta)$ configurations over short training durations (20 epochs) and long training durations (200 epochs) for VGG11 with weight decay $\lambda = 0.01$. Our criteria (short and long training duration) are chosen to find parameters $(\alpha, \beta)$ that provide a rapid decrease in training loss in the early stages and the best test accuracy for long training duration.

These results highlight a tendency for efficient couples; we choose for further experiments the values $(\alpha, \beta) = (0.1, 0.9)$ which correspond to aggressive optimization of the training loss for short training durations, and $(\alpha, \beta) = (2.0, 2.0)$ which provides very good results for longer training durations. Additional results for VGG11 and ResNet18 with and without weight decay are in Appendix F.4, which are qualitatively similar.

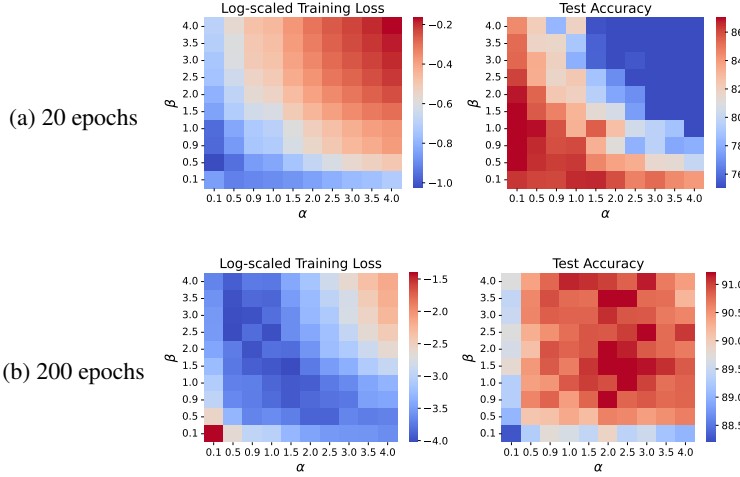

Figure 1: Log-scale training loss and test accuracies for hyperparameters $(\alpha, \beta)$ with VGG11 on CIFAR10 at 20 and 200 epochs. Optimal learning rate $\gamma_0 = 10^{-3}$ and weight decay $\lambda = 0.01$, with one random seed.

**Validation and comparison with AdamW:** We confirm our hyperparameter choices ($\gamma_0 = 10^{-3}$, $\lambda = 0.01$) by reproducing the experiment with 8 random seeds and comparing with AdamW using the same settings. Based on hyperparameter tuning, we select two pairs of $(\alpha, \beta)$ with different training speeds. As shown in Figure 2 (and Appendix F for ResNet18), with $(\alpha, \beta) = (0.1, 0.9)$, INNAprop improves training loss and test accuracy rapidly by the 100th epoch, maintaining the highest training accuracy. With $(\alpha, \beta) = (2.0, 2.0)$, INNAprop trains more slowly but achieves higher final test accuracy. This is aligned with the experiments described in Figure 1. In Table 2, we compare the performance of different networks on CIFAR-10 using INNAprop and AdamW optimizers.

**Remark 4 (Trade-off between fast learning and good generalization)** For CIFAR-10 experiments, INNAprop offers flexibility in adjusting convergence speed through $(\alpha, \beta)$. Faster training configurations generally lead to weaker generalization compared to slower ones, highlighting the trade-off between quick convergence and generalization (Wilson et al., 2017; Zhang et al., 2020).

### 3.2 EXTENSIVE EXPERIMENTS ON LARGE-SCALE VISION MODELS

We present experiments on large-scale vision benchmarks with the hyperparameters of Section 3.1.

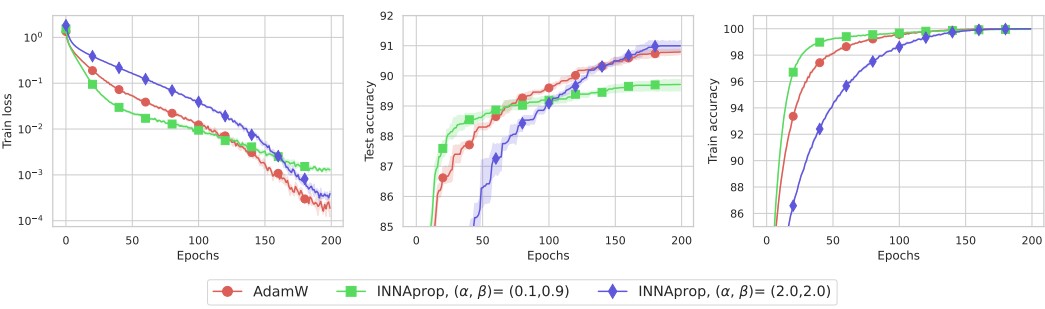

Figure 2: Training VGG11 on CIFAR10. Left: train loss, middle: test accuracy (%), right: train accuracy (%), with 8 random seeds.

Table 2: Test accuracy (%) of ResNet-18, VGG11, and DenseNet121 on CIFAR-10 using AdamW optimized weight decay and learning rate. Results are averaged over eight runs.

| Model | Optimizer | Test accuracy |
|---|---|---|
| Training on CIFAR-10 over 200 epochs | | |
| ResNet18 | AdamW | 91.14 |
| | INNAprop ($\alpha = 2.0, \beta = 2.0$) | **91.58** |
| VGG11 | AdamW | 90.79 |
| | INNAprop ($\alpha = 2.0, \beta = 2.0$) | **90.99** |
| DenseNet121 | AdamW | 86.19 |
| | INNAprop ($\alpha = 0.1, \beta = 0.9$) | **86.91** |

**Resnets on ImageNet:** We consider the larger scale ImageNet-1k benchmark (Krizhevsky et al., 2012). We train a ResNet-18 and a ResNet-50 (He et al., 2016) for 90 epochs with a mini-batch of size of 256 as in Chen et al. (2023b); Zhuang et al. (2020). We used the same cosine scheduler for both AdamW and INNAprop with initial learning rate $\gamma_0 = 10^{-3}$ as reported in Chen et al. (2023b); Zhuang et al. (2020); Chen et al. (2018). The weight decay of AdamW is set to $\lambda = 0.01$ for the ResNet18, instead of $\lambda = 0.05$ reported in Zhuang et al. (2020); Chen et al. (2018) because it improved the test accuracy from 67.93 to 69.43. The results of the ResNet18 experiment are presented in Figure 14 in Appendix F. The figure shows that our algorithm with $(\alpha, \beta) = (0.1, 0.9)$ outperforms AdamW in test accuracy (70.12 vs 69.34), though the training loss decreases faster initially but slows down towards the end of training.

For the ResNet50, we kept the value $\lambda = 0.1$ as reported in Zhuang et al. (2020); Chen et al. (2018). For INNAprop, we tried two weight decay values $\{0.1, 0.01\}$ and selected $\lambda = 0.01$ as it resulted in a faster decrease in training loss. We report the results in Figure 3, illustrating the advantage of INNAprop. As noted in Section 3.1, INNAprop with $(\alpha, \beta) = (0.1, 0.9)$ reduces training loss quickly but has lower test accuracy compared to AdamW or INNAprop with $(\alpha, \beta) = (2.0, 2.0)$. For $(\alpha, \beta) = (2.0, 2.0)$, the loss decrease is similar to AdamW, with no clear advantage for either method. This obviously suggests developing scheduling strategies for damping parameters $(\alpha, \beta)$. This would require a much more computation-intensive tuning, far beyond the numerical resources used in the current work. In Table 3, we present the performance of INNAprop achieved using minimal hyperparameter tuning, as explained in Table 1.

**Vision transformer (ViT) on ImageNet:** We performed the same experiment with a ViT-B/32 architecture over 300 epochs with a mini-batch size of 1024, following Defazio & Mishchenko (2023); Mishchenko & Defazio (2023). For AdamW, we used a cosine scheduler with a linear warmup (30 epochs) and the initial learning rate and weight decay from Defazio & Mishchenko (2023). For INNAprop, we tested weight decay values of $\{0.1, 0.01\}$, selecting $\lambda = 0.1$ for better test accuracy. Results in Figure 3 show the advantage of INNAprop. For faster convergence using INNAprop $(0.1, 0.9)$, we recommend a weight decay of $\lambda = 0.01$ (see Figure 15 in the Appendix).

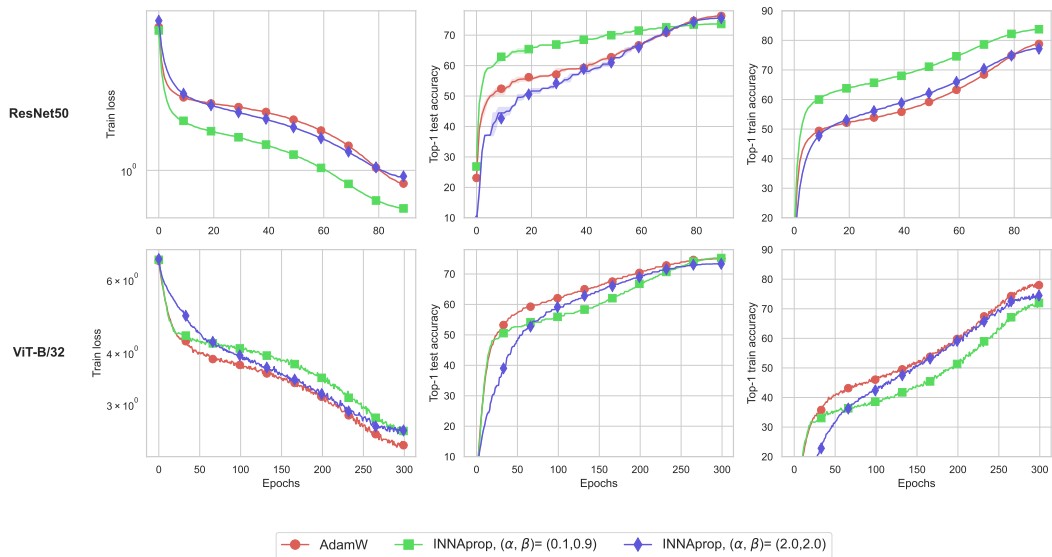

Figure 3: Training a ResNet50 (top) and ViT-B/32 (bottom) on ImageNet. Left: train loss, middle: Top-1 test accuracy (%), right: Top-1 train accuracy (%). 3 random seeds.

In the ImageNet experiments, we evaluated INNAprop for rapid early training and optimal final test accuracy without tuning $(\gamma_0, \alpha, \beta)$. For ViT-B/32 with $\lambda = 0.1$, INNAprop achieved lower training loss and higher final test accuracy than AdamW (75.23 vs. 75.02).

Table 3: Top-1 and Top-5 accuracy (%) of ResNet-18, ResNet-50, and ViT-B/32 on ImageNet. Results are averaged from three runs for ResNets and one run for ViT-B/32. AdamW favored as in Table 1.

| Model | Optimizer | Top-1 accuracy | Top-5 accuracy |
|---|---|---|---|
| | Train from scratch on ImageNet | | |
| ResNet18 | AdamW | 69.34 | 88.71 |
| | INNAprop ($\alpha = 0.1, \beta = 0.9$) | **70.12** | **89.21** |
| ResNet50 | AdamW | 76.33 | 93.04 |
| | INNAprop ($\alpha = 1.0, \beta = 1.0$) | **76.43** | **93.15** |
| ViT-B/32 | AdamW | 75.02 | 91.52 |
| | INNAprop ($\alpha = 0.1, \beta = 0.9$) | **75.23** | **91.77** |

**Fintetuning VGG11 and ResNet18 models on Food101:** We fine-tuned ResNet-18 and VGG-11 models on the Food101 dataset (Bossard et al., 2014) for 20 epochs, using pre-trained models on ImageNet-1k. Since weight decay and learning rate values for AdamW were not found in the literature, we chose the default AdamW weight decay value, $\lambda = 0.01$. We used a cosine scheduler and tried one run for each initial learning rate value in $\{10^{-5}, 5 \times 10^{-5}, 10^{-4}, 5 \times 10^{-4}, 10^{-3}\}$. The best result for AdamW was obtained for $\gamma_0 = 10^{-4}$, and we kept the same setting for INNAprop. See for this Figure 4, where INNAprop performs no worse than AdamW on three random seeds.

**Conclusion and recommendation for image classification:** Tuning $(\alpha, \beta)$ significantly impacts training. Based on heatmaps in Section 3.1 and figures in Section 3.2, we recommend using $\alpha = 0.1$ and $\beta \in [0.5, 1.5]$ for shorter training (e.g., fine-tuning). For longer training, $\alpha, \beta \geq 1$ is preferable. In both cases, our algorithm either matches or outperforms AdamW.

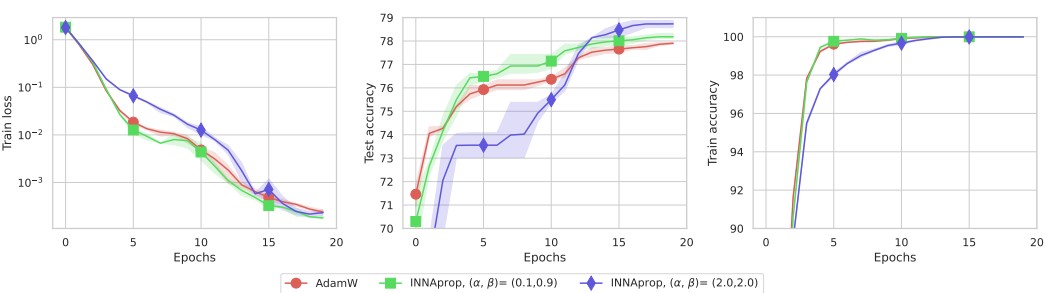

Figure 4: Finetuning a VGG11 on Food101. Left: train loss, middle: test accuracy (%), right: train accuracy (%). Qualitatively similar results for ResNet18 are in Figure 13 in Appendix F. 3 random seeds.

## 3.3 PRE-TRAINING AND FINE-TUNING GPT2

We present experimental results on LLMs using the hyperparameters selected as in Section 3.1.

**Training GPT-2 from scratch:** We train various GPT-2 transformer models from scratch (Radford et al., 2019) using the nanoGPT repository[2] on the OpenWebText dataset. For all models, gradients are clipped to a norm of 1, following Mishchenko & Defazio (2023); Liu et al. (2023); Brown et al. (2020). We use AdamW with hyperparameters from the literature (Liu et al., 2023; Brown et al., 2020), the standard configuration for LLM pre-training. The reported RMSprop parameter $\beta_2 = 0.95$ is different from AdamW's default (0.999), the weight decay is $\lambda = 0.1$ and $\gamma_0$ depending on the network size (see Brown et al. (2020); Liu et al. (2023)). For example, GPT-2 small works with an initial learning rate $\gamma_0 = 6 \times 10^{-4}$. For INNAprop, we keep the same values for $\lambda$ and $\gamma_0$ as AdamW, and use the RMSprop parameter $\sigma = 0.99$ (corresponding to $\beta_2$ for AdamW), which provides the best results among values $\{0.9, 0.95, 0.99\}$ on GPT-2 mini. We use this setting for all our GPT-2 experiments with $(\alpha, \beta) = (0.1, 0.9)$. The results are in Figure 5. INNAprop leads to a faster decrease in validation loss during the early stages compared to the baseline for GPT-2 models of Mini (30M), Small (125M), and Medium (355M) sizes. Its performance could be further improved with more thorough tuning of hyperparameters $(\alpha, \beta, \sigma, \lambda)$. For GPT-2 small, we also include a comparison with Sophia-G, using the hyperparameters provided in the literature [3] (Liu et al., 2023).

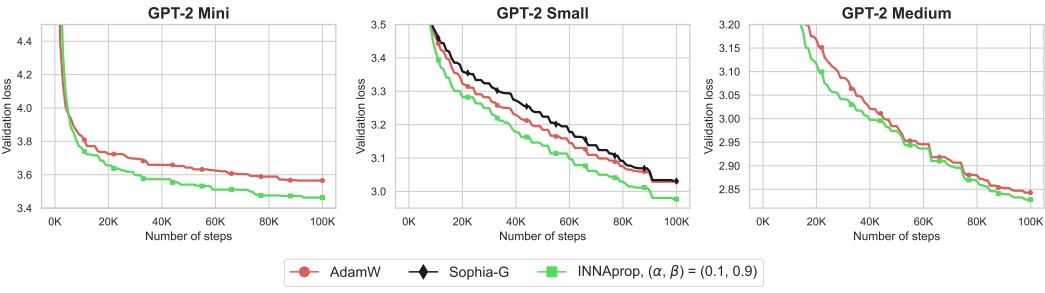

Figure 5: GPT-2 training from scratch on OpenWebText: Sophia-G excluded for mini due to lack of recommendations; medium case failed with author-suggested settings.

**Fine-tune GPT-2 with LoRA:** Using LoRA (Hu et al., 2021), we fine-tune the same GPT-2 models on the E2E dataset, consisting of roughly 42000 training 4600 validation, and 4600 test examples from the restauration domain. We compare AdamW and INNAprop for 5 epochs, as recommended in Hu et al. (2021). We use for both algorithms the same linear learning rate schedule, the recommended mini-batch size, and the RMSprop parameter ($\beta_2 = \sigma = 0.999$); these are listed in Table 11 in Hu et al. (2021). The results are displayed in Figure 6 and Table 4, where we see the perplexity

---

[2]https://github.com/karpathy/nanoGPT
[3]https://github.com/Liuhong99/Sophia

mean result over 3 random seeds. INNAprop with $(\alpha, \beta) = (0.1, 0.9)$ consistently achieves lower perplexity loss compared to AdamW across all GPT-2 fine-tuning experiments.

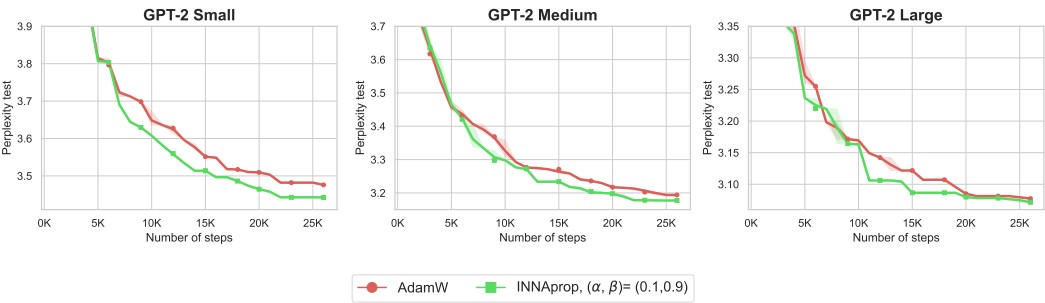

Figure 6: Perplexity test with GPT-2 E2E Dataset with LoRA finetuning on five epochs. Three random seeds.

We synthetize the performance of our algorithm on LLMs below and we emphasize the capabilities of INNAprop compared to AdamW in the context of early training where gains are considerable.

Table 4: Performance comparison for GPT-2 training from scratch on OpenWebText (validation loss) and fine-tuning with LoRA on the E2E dataset (perplexity).

| Model | AdamW best | INNAprop best | Steps to match AdamW |
|---|---|---|---|
| **GPT-2 Training from scratch (Validation loss)** | | | |
| GPT-2 mini | 3.57 | **3.47** | 51,000 (1.96× faster) |
| GPT-2 small | 3.03 | **2.98** | 79,000 (1.26× faster) |
| GPT-2 medium | 2.85 | **2.82** | 83,000 (1.2× faster) |
| **GPT-2 with LoRA (Perplexity test)** | | | |
| GPT-2 small | 3.48 | **3.44** | 19,000 (1.31× faster) |
| GPT-2 medium | 3.20 | **3.17** | 20,000 (1.25× faster) |
| GPT-2 large | 3.09 | **3.06** | 20,000 (1.25× faster) |

## 4 CONCLUSION

INNAprop is an optimizer that leverages second-order geometric information while maintaining memory and computational footprints similar to AdamW. Experiments on text modeling and image classification show that INNAprop consistently matches or exceeds AdamW's performance.

We systematically favored AdamW through the choice of recommended hyperparameters (schedulers, learning rates, weight decay). Hyperparameter tuning for friction parameters $(\alpha, \beta)$ was conducted using a grid search on CIFAR-10 (see Figure 18). Further experiments in that direction could greatly improve the efficiency of INNAprop.

For language models, INNAprop with $(\alpha, \beta) = (0.1, 0.9)$ performs consistently well across all training durations, both for pre-training from scratch and for fine-tuning. We recommend that value for LLMs. Early training achieves notable successes (refer to Table 4).

In image classification, $(\alpha, \beta) = (0.1, 0.9)$ accelerates short-term learning, while higher values like $(\alpha, \beta) = (2.0, 2.0)$ improve test accuracy during longer training runs. Moreover, $(\alpha, \beta) = (2.0, 2.0)$ is effective for fine-tuning, offering a good balance between convergence speed and final accuracy.

These experiments illustrate consistent performances of the proposed method over a diversity of benchmarks, architecture, and model scales, making INNAprop a promising competitor for the training of large neural networks. Future research will be focused on the design of schedulers for the hyperparameters $\alpha$ and $\beta$.

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

This is the appendix for "A second-order-like optimizer with adaptive gradient scaling for deep learning".

CONTENTS

## A  A REMINDER ON OPTIMIZATION ALGORITHMS

Considering the problem in Equation (1) and setting $\nabla \mathcal{J}(\theta_k) = g_k$, we outline several well-known update rule optimizers.

Table 5: Update rules considered for known optimizers. SGD is due to (Robbins & Monro, 1951), Momentum to (Polyak, 1964), Nesterov to (Nesterov, 1983), RMSprop + Momentum to (Graves, 2013), Adam to (Kingma & Ba, 2014), NAdam to (Dozat, 2016) and INNA to (Castera et al., 2021).

$\underline{\text{SGD}(\gamma_k)}$

$\theta_{k+1} = \theta_k - \gamma_k g_k$

$\underline{\text{Adam}(\gamma_k, \beta_1, \beta_2, \epsilon)}$

$m_0 = 0, v_0 = 0$

$m_{k+1} = \beta_1 m_k + (1 - \beta_1) g_k$

$v_{k+1} = \beta_2 v_k + (1 - \beta_2) g_k^2$

$\theta_{k+1} = \theta_k - \gamma_k \dfrac{m_{k+1}}{\sqrt{v_{k+1}} + \epsilon}$

$\underline{\text{NAdam}(\gamma_k, \psi, \beta_1, \beta_2, \epsilon)}$

$m_0 = 0, v_0 = 0$

$\mu_k = \beta_1 (1 - \dfrac{1}{2} 0.96^{k\psi})$

$m_{k+1} = \beta_1 m_k + (1 - \beta_1) g_k$

$v_{k+1} = \beta_2 v_k + (1 - \beta_2) g_k^2$

$\theta_{k+1} = \theta_k - \gamma_k \dfrac{\mu_{k+1} m_{k+1} + (1 - \mu_k) g_k}{\sqrt{v_{k+1}} + \epsilon}$

$\underline{\text{Momentum}(\gamma_k, \beta_1)}$

$v_0 = 0$

$v_{k+1} = \beta_1 v_k + (1 - \beta_1) g_k$

$\theta_{k+1} = \theta_k - \gamma_k v_{k+1}$

$\underline{\text{RMSprop + Momentum}(\gamma_k, \beta_1, \beta_2, \epsilon)}$

$v_0 = 1, m_0 = 0$

$v_{k+1} = \beta_2 v_k + (1 - \beta_2) g_k^2$

$m_{k+1} = \beta_1 m_k + \dfrac{g_k}{\sqrt{v_{k+1}} + \epsilon}$

$\theta_{k+1} = \theta_k - \gamma_k m_{k+1}$

$\underline{\text{INNA}(\gamma_k, \alpha, \beta)}$

$\psi_0 = (1 - \alpha\beta) \theta_0$

$\psi_{k+1} = \psi_k + \gamma_k \left( (\dfrac{1}{\beta} - \alpha) \theta_k - \dfrac{1}{\beta} \psi_k \right)$

$\theta_{k+1} = \theta_k + \gamma_k \left( (\dfrac{1}{\beta} - \alpha) \theta_k - \dfrac{1}{\beta} \psi_k - \beta g_k \right)$

## B  DERIVATION OF INNAPROP FROM DIN

We consider (8) which was a discretization of (6), namely:

$$v_{k+1} = \sigma_2 v_k + (1 - \sigma_2) g_k^2 \tag{9}$$

$$\frac{\theta_{k+1} - 2\theta_k + \theta_{k-1}}{\gamma^2} + \alpha \frac{\theta_k - \theta_{k-1}}{\gamma} + \beta \frac{\frac{g_k}{\sqrt{v_{k+1}}+\epsilon} - \frac{g_{k-1}}{\sqrt{v_k}+\epsilon}}{\gamma} + \frac{g_{k-1}}{\sqrt{v_k} + \epsilon} = 0. \tag{10}$$

This gives

$$\frac{1}{\gamma}\left(\left(\frac{\theta_{k+1} - \theta_k}{\gamma} + \beta\frac{g_k}{\sqrt{v_{k+1}} + \epsilon}\right) - \left(\frac{\theta_k - \theta_{k-1}}{\gamma} + \beta\frac{g_{k-1}}{\sqrt{v_k} + \epsilon}\right)\right) = -\alpha\frac{\theta_k - \theta_{k-1}}{\gamma} - \frac{g_{k-1}}{\sqrt{v_k} + \epsilon}$$

and thus

$$\frac{1}{\gamma}\left(\left(\frac{\theta_{k+1} - \theta_k}{\gamma} + \beta\frac{g_k}{\sqrt{v_{k+1}} + \epsilon}\right) - \left(\frac{\theta_k - \theta_{k-1}}{\gamma} + \beta\frac{g_{k-1}}{\sqrt{v_k} + \epsilon}\right)\right)$$
$$= \left(\frac{1}{\beta} - \alpha\right)\frac{\theta_k - \theta_{k-1}}{\gamma} - \frac{1}{\beta}\left(\frac{\theta_k - \theta_{k-1}}{\gamma} + \beta\frac{g_{k-1}}{\sqrt{v_k} + \epsilon}\right).$$

Multiplying by $\beta$, we obtain

$$\frac{1}{\gamma}\left(\left(\beta\frac{\theta_{k+1} - \theta_k}{\gamma} + \beta^2\frac{g_k}{\sqrt{v_{k+1}} + \epsilon}\right) - \left(\beta\frac{\theta_k - \theta_{k-1}}{\gamma} + \beta^2\frac{g_{k-1}}{\sqrt{v_k} + \epsilon}\right)\right)$$
$$= (1 - \alpha\beta)\frac{\theta_k - \theta_{k-1}}{\gamma} - \frac{\theta_k - \theta_{k-1}}{\gamma} - \beta\frac{g_{k-1}}{\sqrt{v_k} + \epsilon}$$

after rearranging all terms

$$\frac{1}{\gamma}\left(\left(\beta\frac{\theta_{k+1} - \theta_k}{\gamma} + \beta^2\frac{g_k}{\sqrt{v_{k+1}} + \epsilon} + (\alpha\beta - 1)\theta_k\right) - \left(\beta\frac{\theta_k - \theta_{k-1}}{\gamma} + \beta^2\frac{g_{k-1}}{\sqrt{v_k} + \epsilon} + (\alpha\beta - 1)\theta_{k-1}\right)\right)$$
$$= -\frac{\theta_k - \theta_{k-1}}{\gamma} - \beta\frac{g_{k-1}}{\sqrt{v_k} + \epsilon}\,.$$

Setting $\psi_{k-1} = -\beta\frac{\theta_k - \theta_{k-1}}{\gamma} - \beta^2\frac{g_{k-1}}{\sqrt{v_k}+\epsilon} - (\alpha\beta - 1)\theta_{k-1}$, we obtain the recursion

$$v_{k+1} = \sigma_2 v_k + (1 - \sigma_2)g_k^2 \tag{11}$$

$$\frac{\psi_k - \psi_{k-1}}{\gamma} = -\frac{\psi_{k-1}}{\beta} - \left(\alpha - \frac{1}{\beta}\right)\theta_{k-1} \tag{12}$$

$$\frac{\theta_{k+1} - \theta_k}{\gamma} = \frac{-1}{\beta}\psi_k - \beta\frac{g_k}{\sqrt{v_{k+1}} + \epsilon} - \left(\alpha - \frac{1}{\beta}\right)\theta_k \tag{13}$$

We can also rewrite the above as follows:

$$v_{k+1} = \sigma_2 v_k + (1 - \sigma_2)g_k^2$$
$$\psi_{k+1} = \psi_k\left(1 - \frac{\gamma}{\beta}\right) + \gamma\left(\frac{1}{\beta} - \alpha\right)\theta_k,$$
$$\theta_{k+1} = \theta_k\left(1 + \gamma\left(\frac{1}{\beta} - \alpha\right)\right) - \frac{\gamma}{\beta}\psi_k - \gamma\beta\frac{g_k}{\sqrt{v_{k+1}} + \epsilon}\,.$$

We can save a memory slot by avoiding the storage of $\psi_k$:

$$\psi_{k+1} = \psi_k \left(1 - \frac{\gamma}{\beta}\right) + \gamma \left(\frac{1}{\beta} - \alpha\right) \theta_k, \tag{14}$$

$$\Leftrightarrow \quad \psi_k = \frac{\beta}{\beta - \gamma} \left(\psi_{k+1} - \gamma \left(\frac{1}{\beta} - \alpha\right) \theta_k\right) = \frac{\beta}{\beta - \gamma}\psi_{k+1} - \frac{\beta}{\beta - \gamma}\gamma \left(\frac{1}{\beta} - \alpha\right) \theta_k$$

$$\theta_{k+1} = \theta_k \left(1 + \gamma \left(\frac{1}{\beta} - \alpha\right)\right) - \frac{\gamma}{\beta}\psi_k - \gamma\beta \frac{g_k}{\sqrt{v_{k+1}} + \epsilon}$$

$$= \theta_k + \gamma \left(\frac{1}{\beta} - \alpha\right) \theta_k - \frac{\gamma}{\beta - \gamma}\psi_{k+1} + \frac{\gamma}{\beta - \gamma}\gamma \left(\frac{1}{\beta} - \alpha\right) \theta_k - \gamma\beta \frac{g_k}{\sqrt{v_{k+1}} + \epsilon}$$

$$= \theta_k + \left(1 + \frac{\gamma}{\beta - \gamma}\right)\gamma \left(\frac{1}{\beta} - \alpha\right) \theta_k - \frac{\gamma}{\beta - \gamma}\psi_{k+1} - \gamma\beta \frac{g_k}{\sqrt{v_{k+1}} + \epsilon}$$

$$= \theta_k + \left(\frac{\beta}{\beta - \gamma}\right)\gamma \left(\frac{1}{\beta} - \alpha\right) \theta_k - \frac{\gamma}{\beta - \gamma}\psi_{k+1} - \gamma\beta \frac{g_k}{\sqrt{v_{k+1}} + \epsilon}$$

$$= \theta_k + \left(\frac{\gamma(1 - \beta\alpha)}{\beta - \gamma}\right) \theta_k - \frac{\gamma}{\beta - \gamma}\psi_{k+1} - \gamma\beta \frac{g_k}{\sqrt{v_{k+1}} + \epsilon}$$

$$= \left(1 + \frac{\gamma(1 - \beta\alpha)}{\beta - \gamma}\right) \theta_k - \frac{\gamma}{\beta - \gamma}\psi_{k+1} - \gamma\beta \frac{g_k}{\sqrt{v_{k+1}} + \epsilon} \tag{15}$$

Finally, we merely need to use 3 memory slots having the underlying dimension size $p$:

$$v_{k+1} = \sigma_2 v_k + (1 - \sigma_2)g_k^2$$

$$\psi_{k+1} = \psi_k \left(1 - \frac{\gamma}{\beta}\right) + \gamma \left(\frac{1}{\beta} - \alpha\right) \theta_k,$$

$$\theta_{k+1} = \left(1 + \frac{\gamma(1 - \beta\alpha)}{\beta - \gamma}\right) \theta_k - \frac{\gamma}{\beta - \gamma}\psi_{k+1} - \gamma\beta \frac{g_k}{\sqrt{v_{k+1}} + \epsilon}$$

---

**Algorithm 2** INNAprop

---

1: **Objective function:** $\mathcal{J}(\theta)$ for $\theta \in \mathbb{R}^p$.
2: **Constant step-size:** $\gamma > 0$
3: **Hyper-parameters:** $\sigma \in [0, 1]$, $\alpha \geq 0$, $\beta > \gamma$, $\epsilon = 10^{-8}$.
4: **Initialization:** $\theta_0$, $v_0 = 0$, $\psi_0 = (1 - \alpha\beta)\theta_0$.
5: **for** $k = 1$ **to** K **do**
6: $\quad g_k = \nabla\mathcal{J}(\boldsymbol{\theta}_k)$
7: $\quad \boldsymbol{v}_{k+1} \leftarrow \sigma\boldsymbol{v}_k + (1 - \sigma)\boldsymbol{g}_k^2$
8: $\quad \boldsymbol{\psi}_{k+1} \leftarrow \left(1 - \frac{\gamma}{\beta}\right)\boldsymbol{\psi}_k + \gamma\left(\frac{1}{\beta} - \alpha\right)\boldsymbol{\theta}_k$
9: $\quad \boldsymbol{\theta}_{k+1} \leftarrow \left(1 + \frac{\gamma(1 - \alpha\beta)}{\beta - \gamma}\right)\boldsymbol{\theta}_k - \frac{\gamma}{\beta - \gamma}\boldsymbol{\psi}_{k+1} - \gamma\beta\frac{\boldsymbol{g}_k}{\sqrt{v_{k+1}} + \epsilon}$
10: **return** $\boldsymbol{\theta}_{K+1}$

---

### B.1 EQUIVALENCE BETWEEN A SPECIAL CASE OF INNAPROP AND ADAM WITHOUT MOMENTUM

In this section, we demonstrate that INNAprop with $\alpha = 1$ and $\beta = 1$ is equivalent to Adam (Kingma & Ba, 2014) without momentum ($\beta_1 = 0$). To illustrate this, we analyze the update rules of both algorithms. We assume that the RMSprop parameter $\beta_2$ (for Adam) and $\sigma$ (for INNAprop) are equal. Starting with INNAprop, we initialize $\psi_0 = (1 - \alpha\beta)\theta_0$. For $\alpha = 1$ and $\beta = 1$, this simplifies to $\psi_0 = 0$. The update for $\psi$ becomes:

$$\psi_{k+1} = \left(1 - \frac{\gamma}{\beta}\right)\psi_k + \gamma\left(\frac{1}{\beta} - \alpha\right)\theta_k = (1 - \gamma)\psi_k$$

Given that $\psi_0 = 0$, it follows that $\psi_k = 0$ for all $k$. The parameter update rule for INNAprop is:

$$\theta_{k+1} = \left(1 + \frac{\gamma(1 - \alpha\beta)}{\beta - \gamma}\right)\theta_k - \frac{\gamma}{\beta - \gamma}\psi_{k+1} - \gamma\beta\frac{g_k}{\sqrt{v_{k+1}} + \epsilon}$$

Replacing $\alpha = 1$, $\beta = 1$, and $\psi_k = 0$, we get:

$$\theta_{k+1} = \theta_k - \gamma\frac{g_k}{\sqrt{v_{k+1}} + \epsilon}$$

Here, $g_k$ is the gradient, and $v_{k+1}$ is the exponential moving average of the squared gradients:

$$v_{k+1} = \sigma v_k + (1 - \sigma)g_k^2$$

The Adam optimizer uses two moving averages, $m_k$ (momentum term) and $v_k$ (squared gradients):

$$m_k = \beta_1 m_{k-1} + (1 - \beta_1)g_k$$
$$v_k = \sigma v_{k-1} + (1 - \sigma)g_k^2$$

Setting $\beta_1 = 0$, the momentum term $m_k$ simplifies to $m_k = g_k$. The update rule becomes:

$$\theta_{k+1} = \theta_k - \gamma\frac{g_k}{\sqrt{v_k} + \epsilon}$$

This matches the form of Adam's update rule without the momentum term, confirming that INNAprop with $\alpha = 1$ and $\beta = 1$ is equivalent to Adam with $\beta_1 = 0$.

---

**Algorithm 3** INNAprop with $(\alpha, \beta) = (1, 1)$

---

1: **Objective function:** $\mathcal{J}(\theta)$ for $\theta \in \mathbb{R}^p$.
2: **Constant step-size:** $\gamma > 0$
3: **Hyper-parameters:** $\sigma \in [0, 1]$, $\alpha \geq 0$, $\beta > \gamma$, $\epsilon = 10^{-8}$.
4: **Initialization:** time step $k \leftarrow 0$, parameter vector $\theta_0$, $v_0 = 0$.
5: **repeat**
6:    $k \leftarrow k + 1$
7:    $\boldsymbol{g}_k = \nabla\mathcal{J}(\boldsymbol{\theta}_k)$
8:    $\boldsymbol{v}_{k+1} \leftarrow \sigma\boldsymbol{v}_k + (1 - \sigma)\boldsymbol{g}_k^2$
9:    $\hat{\boldsymbol{v}}_{k+1} \leftarrow \boldsymbol{v}_{k+1}/(1 - \sigma^k)$
10:   $\boldsymbol{\theta}_{k+1} \leftarrow \boldsymbol{\theta}_k - \gamma_k\left(\boldsymbol{g}_k/(\sqrt{\hat{\boldsymbol{v}}_{k+1}} + \epsilon)\right)$
11: **until** *stopping criterion is met*
12: **return** optimized parameters $\boldsymbol{\theta}_{k+1}$

---

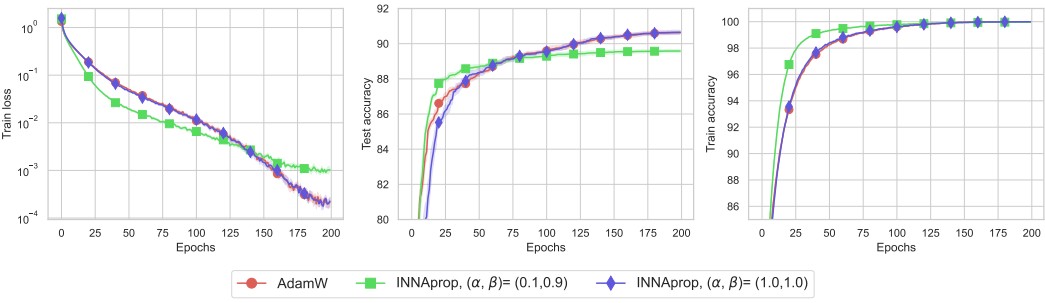

Figure 7: Training VGG11 on CIFAR10. Left: train loss, middle: test accuracy (%), right: train accuracy (%), with 8 random seeds.

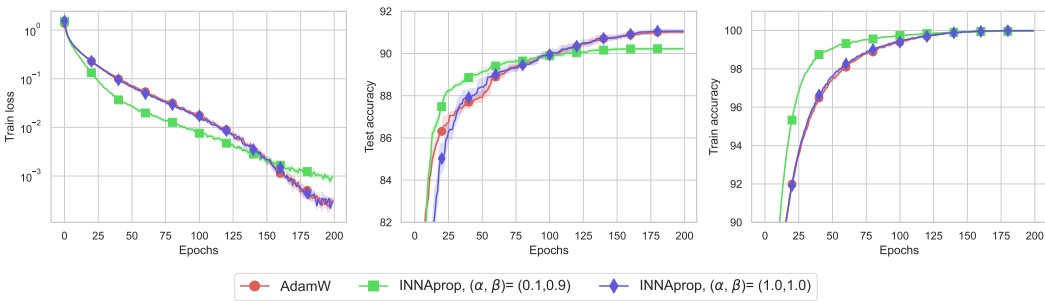

Figure 8: Training ResNet18 on CIFAR10. Left: train loss, middle: test accuracy (%), right: train accuracy (%), with 8 random seeds.

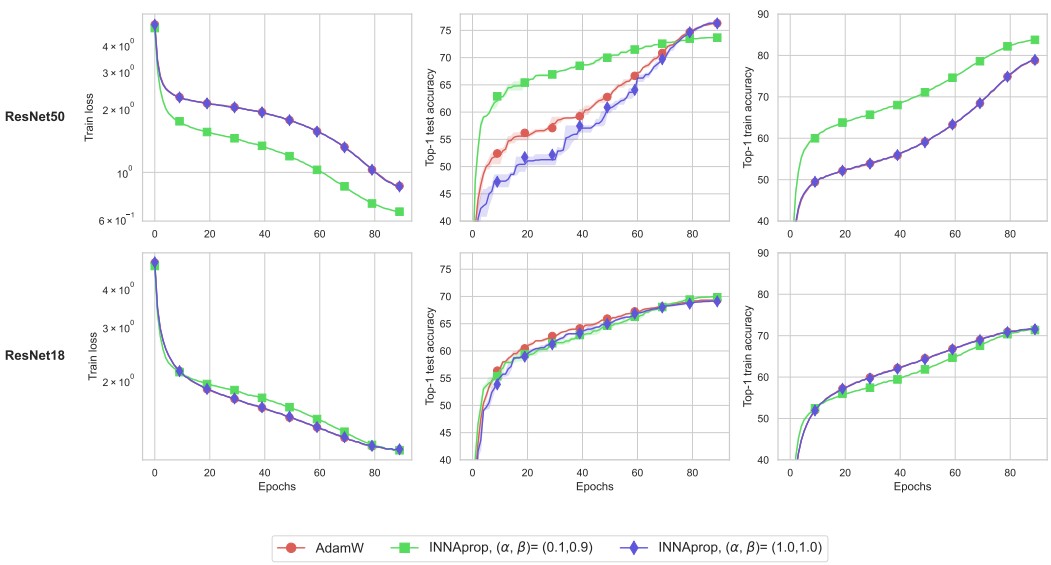

Figure 9: Training a ResNet50 (top) and ResNet18 (bottom) on ImageNet. Left: train loss, middle: Top-1 test accuracy (%), right: Top-1 train accuracy (%). 3 random seeds.

## C  ALTERNATIVE DISCRETIZATIONS

### C.1  AN ALTERNATIVE DERIVATION OF INNAPROP

As mentioned in Remark 2, we can obtain INNAprop easily from INNA (Castera et al., 2021). The algorithm INNA writes (see Table 5):

$$\psi_{k+1} = \psi_k + \gamma_k \left( (\frac{1}{\beta} - \alpha)\theta_k - \frac{1}{\beta}\psi_k \right)$$

$$\theta_{k+1} = \theta_k + \gamma_k \left( (\frac{1}{\beta} - \alpha)\theta_k - \frac{1}{\beta}\psi_k - \beta g_k \right)$$

Rearranging the terms and saving a memory slot — use $\psi_{k+1}$ in the second equation instead of $\psi_k$, (see Equation (15) for details)— yields

$$\psi_{k+1} = \psi_k \left( 1 - \frac{\gamma}{\beta} \right) + \gamma \left( \frac{1}{\beta} - \alpha \right) \theta_k$$

$$\theta_{k+1} = \left( 1 + \frac{\gamma(1 - \beta\alpha)}{\beta - \gamma} \right) \theta_k - \frac{\gamma}{\beta - \gamma}\psi_{k+1} - \gamma\beta g_k$$

Now, use the RMSprop proxy directly within INNA. Using the usual RMSprop constants $\sigma \in [0, 1]$ and $\epsilon > 0$, we obtain:

$$v_{k+1} = \sigma v_k + (1 - \sigma)g_k^2$$

$$\psi_{k+1} = \psi_k \left(1 - \frac{\gamma}{\beta}\right) + \gamma \left(\frac{1}{\beta} - \alpha\right) \theta_k$$

$$\theta_{k+1} = \left(1 + \frac{\gamma(1 - \beta\alpha)}{\beta - \gamma}\right) \theta_k - \frac{\gamma}{\beta - \gamma}\psi_{k+1} - \gamma\beta\frac{g_k}{\sqrt{v_{k+1}} + \epsilon}$$

This is INNAprop and the derivation is much more direct, although less illustrative of the geometric features.

## C.2   A VARIANT OF INNAPROP WITH MOMENTUM

**The algorithm.**   We follow the rationale behind the algorithm RMSprop with momentum (Graves, 2013). We therefore start with Equation (8) using the RMSprop proxy for the gradient:

$$v_{k+1} = \sigma v_k + (1 - \sigma)g_k^2$$

$$\frac{\theta_{k+1} - 2\theta_k + \theta_{k-1}}{\gamma} + \alpha\frac{\theta_k - \theta_{k-1}}{\gamma} + \beta\frac{\frac{g_k}{\sqrt{v_{k+1}}+\epsilon} - \frac{g_{k-1}}{\sqrt{v_k}+\epsilon}}{\gamma} + \frac{g_{k-1}}{\sqrt{v_k} + \epsilon} = 0.$$

Rearranging terms, we have

$$v_{k+1} = \sigma v_k + (1 - \sigma)g_k^2$$

$$\theta_{k+1} = \theta_k + (1 - \alpha\gamma)(\theta_k - \theta_{k-1}) - \beta\gamma\left(\frac{g_k}{\sqrt{v_{k+1}} + \epsilon} - \frac{g_{k-1}}{\sqrt{v_k} + \epsilon}\right) - \gamma^2\frac{g_{k-1}}{\sqrt{v_k} + \epsilon}$$

Let us introduce a momentum variable $m_k = \theta_{k-1} - \theta_k$ to obtain:

$$v_{k+1} = \sigma v_k + (1 - \sigma)g_k^2 \tag{16}$$

$$m_{k+1} = (1 - \alpha\gamma)m_k + \gamma^2\frac{g_{k-1}}{\sqrt{v_k} + \epsilon} + \beta\gamma\left(\frac{g_k}{\sqrt{v_{k+1}} + \epsilon} - \frac{g_{k-1}}{\sqrt{v_k} + \epsilon}\right) \tag{17}$$

$$\theta_{k+1} = \theta_k - m_{k+1} \tag{18}$$

As previously need now to optimize the dynamics in terms of storage. For this we rewrite Equation (17) as

$$m_{k+1} = am_k + bg_k - cg_{k-1}. \tag{19}$$

where $a = (1 - \alpha\gamma)$, $b = \beta\gamma$ and $c = \gamma(\beta - \gamma)$. Writing $\tilde{m}_k = m_k - \frac{c}{a}g_{k-1}$, we have

$$\tilde{m}_{k+1} = m_{k+1} - \frac{c}{a}g_k$$

$$= am_k + bg_k - cg_{k-1} - \frac{c}{a}g_k$$

$$= a\left(m_k - \frac{c}{a}g_{k-1}\right) + \left(b - \frac{c}{a}\right)g_k$$

$$= a\tilde{m}_k + \left(b - \frac{c}{a}\right)g_k.$$

Therefore, using this identity, we may rewrite the following

$$m_{k+1} = am_k + bg_k - cg_{k-1},$$

$$\theta_{k+1} = \theta_k - m_{k+1}$$

as

$$\tilde{m}_{k+1} = a\tilde{m}_k + \left(b - \frac{c}{a}\right)g_k,$$

$$\theta_{k+1} = \theta_k - \tilde{m}_{k+1} - \frac{c}{a}g_k.$$

Recalling that $a = (1 - \alpha\gamma)$, $b = \beta\gamma$ and $c = \gamma(\beta - \gamma)$. Finally, we get the following recursion which is an alternative way to integrate RMSprop to INNA:

$$v_{k+1} = \sigma v_k + (1 - \sigma)g_k^2 \tag{20}$$

$$\tilde{m}_{k+1} = (1 - \alpha\gamma)\tilde{m}_k + \gamma^2 \left(\frac{1 - \alpha\beta}{1 - \alpha\gamma}\right) \frac{g_k}{\sqrt{v_{k+1}} + \epsilon} \tag{21}$$

$$\theta_{k+1} = \theta_k - \tilde{m}_{k+1} - \frac{\gamma(\beta - \gamma)}{1 - \alpha\gamma} \frac{g_k}{\sqrt{v_{k+1}} + \epsilon} \tag{22}$$

but as shown below through numerical experiments, the factor $\gamma^2$ is poorly scaled for 32 bits or lower machine precision.

**Numerical experiments.** Using CIFAR-10 dataset, we train a VGG11 network with the momentum version of INNAprop with the hyperparameters $(\alpha, \beta) = (0.1, 0.9)$ above. We used a cosine annealing scheduler with $\gamma_0 = 10^{-3}$ and no weight decay. As seen in Figure 10, the training loss stops decreasing between the 125th and 150th epochs. Upon closely examining the algorithm in this regime, we observe that at the end of training, $\gamma_k^2$ falls below the numerical precision, resulting in unstable behavior in Equation (21).

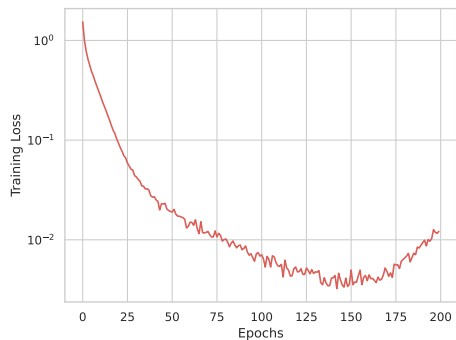

Figure 10: The version of INNA with momentum of Section C.2 is an unstable method.

## C.3 AN APPROACH À LA ADAM

In this section, we mimic the process for deriving Adam from the heavy ball with a RMSprop proxy, see, e.g., Kingma & Ba (2014); Ruder (2016), by simply replacing the heavy ball by DIN[4]. We call this optimizer DINAdam.

From (6), we infer the discretization:

$$\frac{\theta_{k+1} - 2\theta_k + \theta_{k-1}}{\gamma^2} + \alpha\frac{\theta_{k+1} - \theta_k}{\gamma} + \beta\frac{g_k - g_{k-1}}{\gamma} + g_k = 0. \tag{23}$$

Rearranging terms, we have

$$\theta_{k+1} = \theta_k - \frac{\gamma^2}{1 + \alpha\gamma}g_k + \frac{1}{1 + \alpha\gamma}(\theta_k - \theta_{k-1}) - \frac{\beta\gamma}{(1 + \alpha\gamma)}(g_k - g_{k-1}) \tag{24}$$

By introducing the new variable $m_k = (\theta_{k-1} - \theta_k)/\eta$ and setting $\eta > 0$, we can rewrite equation (24) as:

$$m_{k+1} = \frac{1}{(1 + \alpha\gamma)}m_k + \frac{\gamma^2}{(1 + \alpha\gamma)\eta}g_k + \frac{\beta\gamma}{(1 + \alpha\gamma)\eta}(g_k - g_{k-1}) \tag{25}$$

$$\theta_{k+1} = \theta_k - \eta m_{k+1} \tag{26}$$

---

[4]Note that DIN with $\beta = 0$ boils down to the heavy ball method.

To follow the Adam spirit, we set $\sigma_1 = \frac{1}{(1+\alpha\gamma)}$ and $(1 - \sigma_1) = \frac{\gamma^2}{(1+\alpha\gamma)\eta}$. Solving for $\gamma$, we get

$$\frac{\alpha\gamma}{1 + \alpha\gamma} = \frac{\gamma^2}{(1 + \alpha\gamma)\eta} \Rightarrow \gamma = \frac{\eta}{\alpha}$$

Then, we find the following recursion:

$$m_{k+1} = \sigma_1 m_k + (1 - \sigma_1)g_k + \beta\alpha\sigma_1(g_k - g_{k-1}) \tag{27}$$

$$\theta_{k+1} = \theta_k - \eta m_{k+1} \tag{28}$$

From Equation (27), we make a change of variable $\tilde{m}_k = m_k - \alpha\beta g_{k-1}$ to save a memory cell.

$$\tilde{m}_{k+1} = \sigma_1 \tilde{m}_k + (1 - \sigma_1 + \beta\alpha\sigma_1 - \beta\alpha)g_k \tag{29}$$

$$\theta_{k+1} = \theta_k - \eta(\tilde{m}_{k+1} - \alpha\beta g_k) \tag{30}$$

Using the usual RMSprop constants $\sigma_2 \in [0, 1]$ and $\epsilon > 0$, we obtain:

$$v_{k+1} = \sigma_2 v_k + (1 - \sigma_2)g_k^2 \tag{31}$$

$$\tilde{m}_{k+1} = \sigma_1 \tilde{m}_k + (1 - \sigma_1 + \beta\alpha\sigma_1 - \beta\alpha)g_k \tag{32}$$

$$\theta_{k+1} = \theta_k - \eta\frac{\tilde{m}_{k+1} - \alpha\beta g_k}{\sqrt{v_{k+1}} + \epsilon} \tag{33}$$

---

**Algorithm 4** DINAdam

---

1: **Objective function:** $\mathcal{J}(\theta)$ for $\theta \in \mathbb{R}^p$.
2: **Constant step-size:** $\gamma > 0$
3: **Hyper-parameters:** $(\sigma_1, \sigma_2) \in [0, 1]^2$, $\alpha, \beta > 0$, $\epsilon = 10^{-8}$.
4: **Initialization:** $\theta_0$, $v_0 = 0$, $\tilde{m}_0 = 0$.
5: **repeat**
6:     $\boldsymbol{g}_k = \nabla\mathcal{J}(\boldsymbol{\theta}_k)$
7:     $\boldsymbol{v}_{k+1} \leftarrow \sigma_2\boldsymbol{v}_k + (1 - \sigma_2)\boldsymbol{g}_k^2$
8:     $\tilde{\boldsymbol{m}}_{k+1} \leftarrow \sigma_1\tilde{\boldsymbol{m}}_k + (1 - \sigma_1 + \beta\alpha\sigma_1 - \beta\alpha)\boldsymbol{g}_k$
9:     $\boldsymbol{\theta}_{k+1} \leftarrow \boldsymbol{\theta}_k - \gamma\frac{\tilde{\boldsymbol{m}}_{k+1} - \alpha\beta\boldsymbol{g}_k}{\sqrt{\boldsymbol{v}_{k+1}} + \epsilon}$
10: **until** *stopping criterion is met*
11: **return** optimized parameters $\boldsymbol{\theta}_k$

---

**Remark 5** The way RMSprop is added in INNAprop and DINAdam is different. In INNAprop, RMSprop is incorporated directly during the discretization process of Equation (8) for all gradients. However, in DINAdam, RMSprop is added only at the last step, as shown in Equation (31), and only on the gradient in the $\theta_{k+1}$ update. This is how RMSprop was combined with heavy ball to obtain Adam.

**Remark 6** After setting $\alpha = 1$ and $\beta = 0$, we obtain Adam update rules. If $\beta \neq 0$, DINAdam is very close to NAdam algorithm. Hence, we did not investigate this algorithm numerically.

## D   SCHEDULER PROCEDURES

**Cosine annealing (Loshchilov & Hutter, 2016).** Let $\gamma_k$ represent the learning rate at iteration $k$, $T_{\max}$ be the maximum number of iterations (or epochs), and $\gamma_{\min}$ be the minimum learning rate (default value is 0). The learning rate $\gamma_k$ at iteration $k$ is given by:

$$\gamma_k = \gamma_{\min} + \frac{1}{2}(\gamma_0 - \gamma_{\min})\left(1 + \cos\left(\frac{k}{T_{\max}}\pi\right)\right)$$

This scheduler was employed in all image classification experiments except for ViT.

**Cosine annealing with linear warmup (Radford et al., 2018).** Let $\gamma_k$ represent the learning rate at iteration $k$, $\gamma_{\min}$ the minimum learning rate, $\gamma_0$ the initial learning rate, $T_{\text{warmup}}$ the number of iterations for the warmup phase, and $T_{\text{decay}}$ the iteration number after which the learning rate decays to $\gamma_{\min}$. The learning rate is defined as follows:

$$
\gamma_k = \begin{cases} \gamma_0 \cdot \frac{k}{T_{\text{warmup}}}, & \text{if } k < T_{\text{warmup}} \\ \gamma_{\min} + \frac{1}{2} (\gamma_0 - \gamma_{\min}) \left( 1 + \cos \left( \pi \cdot \frac{k - T_{\text{warmup}}}{T_{\text{decay}} - T_{\text{warmup}}} \right) \right), & \text{if } T_{\text{warmup}} \leq k \leq T_{\text{decay}} \\ \gamma_{\min}, & \text{if } k > T_{\text{decay}} \end{cases}
$$

This scheduler was applied in experiments involving training GPT-2 from scratch and for ViT.

**Linear schedule with linear warmup (Hu et al., 2021).** Let $\gamma_k$ represent the learning rate at iteration $k$ and $T_{\max}$ be the maximum number of iterations, $T_{\text{warmup}}$ be the number of warmup steps, and $\gamma_{\min}$ be the minimum learning rate after warmup (default value is typically set to the initial learning rate, $\gamma_0$). The learning rate $\gamma_k$ at iteration $k$ is given by:

$$
\gamma_k = \begin{cases} \gamma_0 \cdot \frac{k}{T_{\text{warmup}}} & \text{if } k < T_{\text{warmup}}, \\ \gamma_0 \cdot \left( 1 - \frac{k - T_{\text{warmup}}}{T_{\max} - T_{\text{warmup}}} \right) & \text{otherwise.} \end{cases}
$$

This scheduler was used for fine-tuning GPT-2 with LoRA.

# E    CHOOSING HYPERPARAMETERS $\alpha$ AND $\beta$ FOR INNAPROP

## E.1    COMPARISON WITH ADAMW

For VGG and ResNet training on CIFAR10, the literature suggest using initial learning rate $\gamma_0 = 10^{-3}$ with a learning rate schedule (Mishchenko & Defazio, 2023; Defazio & Mishchenko, 2023; Yao et al., 2021; Zhuang et al., 2020). Our experiment fix a cosine scheduler where $T_{\max} = 200$ and $\gamma_{\min} = 0$ as it achieves a strong baseline for AdamW (Loshchilov & Hutter, 2016; Mishchenko & Defazio, 2023). We set weight decay $\lambda = 0.1$. Then, we tune the initial learning rate $\gamma_0$ among $\{10^{-4}, 5 \times 10^{-4}, 10^{-3}, 5 \times 10^{-3}, 10^{-2}\}$. In Figure 11, we report the performance in terms of training loss and test accuracy for AdamW. These results confirm the usage of $\gamma_0 = 10^{-3}$.

| (a) Performance rankings with VGG11. | | | (b) Performance rankings with ResNet18. | | |
|---|---|---|---|---|---|
| $\gamma_0$ | **Train loss** | **Test accuracy (%)** | $\gamma_0$ | **Train loss** | **Test accuracy (%)** |
| $10^{-3}$ | 0.00041 | 91.02 | $10^{-3}$ | 0.00040 | 92.1 |
| $5 \times 10^{-3}$ | 0.00047 | 90.86 | $5 \times 10^{-3}$ | 0.00049 | 91.84 |
| $5 \times 10^{-4}$ | 0.00048 | 90.79 | $5 \times 10^{-4}$ | 0.00094 | 92.32 |
| $10^{-2}$ | 0.00057 | 90.41 | $10^{-2}$ | 0.00057 | 90.41 |
| $10^{-4}$ | 0.00081 | 88.49 | $10^{-4}$ | 0.0018 | 87.85 |

Figure 11: Comparative performance of the training loss and test accuracy according to $\gamma_0$. We trained VGG11 and ResNet18 models on CIFAR10 for 200 epochs.

# F ADDITIONAL EXPERIMENTS

## F.1 CIFAR10 EXPERIMENTS

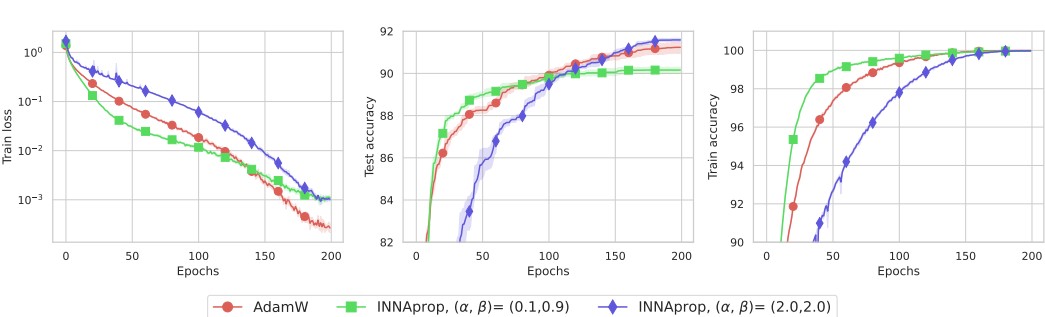

Figure 12: Training ResNet18 on CIFAR10. Left: train loss, middle: test accuracy (%), right: train accuracy (%), with 8 random seeds.

## F.2 FOOD101 EXPERIMENTS

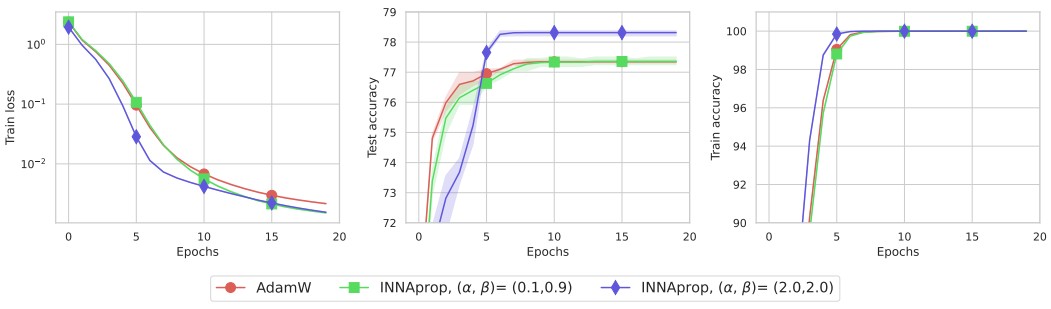

Figure 13: Finetuning a ResNet18 on Food101, same as Figure 4 for ResNet18. Left: train loss, middle: test accuracy (%), right: train accuracy (%), with 3 random seeds.

## F.3 IMAGENET

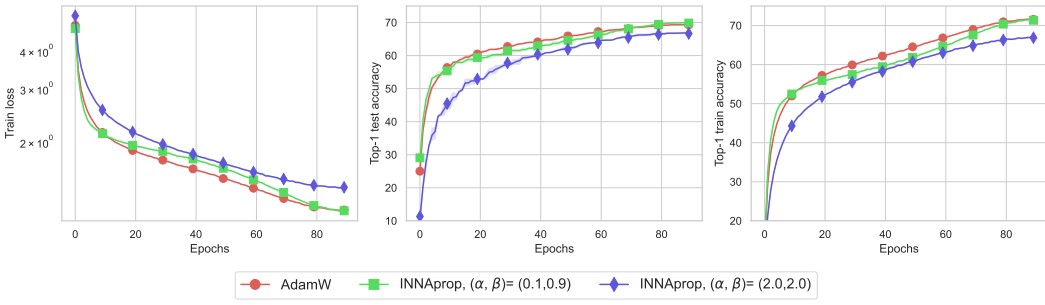

Figure 14: Training ResNet18 on ImageNet. Left: train loss, middle: test accuracy (%), right: train accuracy (%), with 3 random seeds.

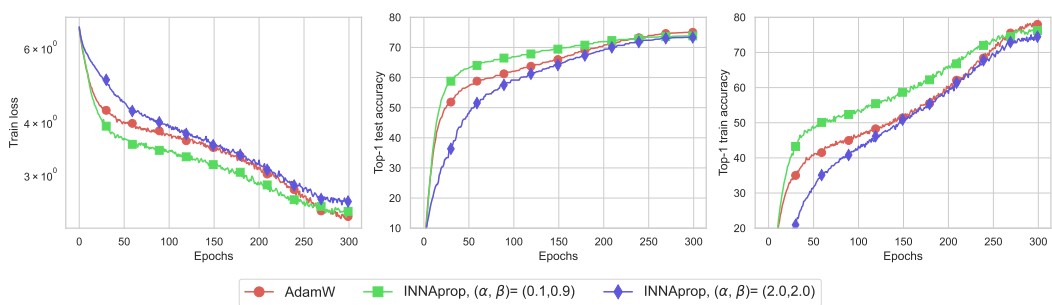

Figure 15: Fast training ViT/B-32 on ImageNet with weight decay $\lambda = 0.01$ for INNAprop $(\alpha, \beta) = (0.1, 0.9)$. Left: train loss, middle: test accuracy (%), right: train accuracy (%), with 3 random seeds.

### F.4 HEATMAP FOR PRELIMINARY TUNING OF $\alpha$ AND $\beta$

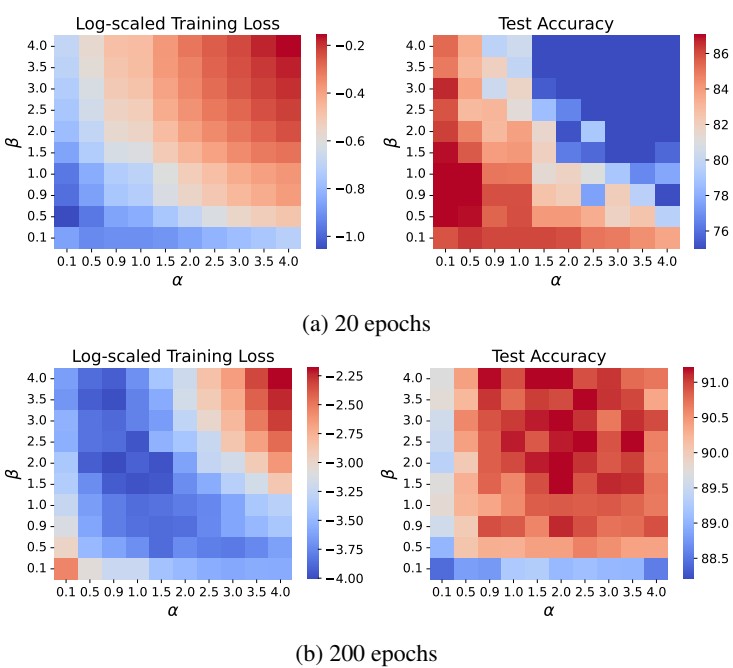

Figure 16: Log-scale training loss and test accuracies for $(\alpha, \beta)$ hyperparameters with VGG11 on CIFAR10 at different epochs. Optimal learning rate $\gamma_0 = 10^{-3}$, weight decay $\lambda = 0$.

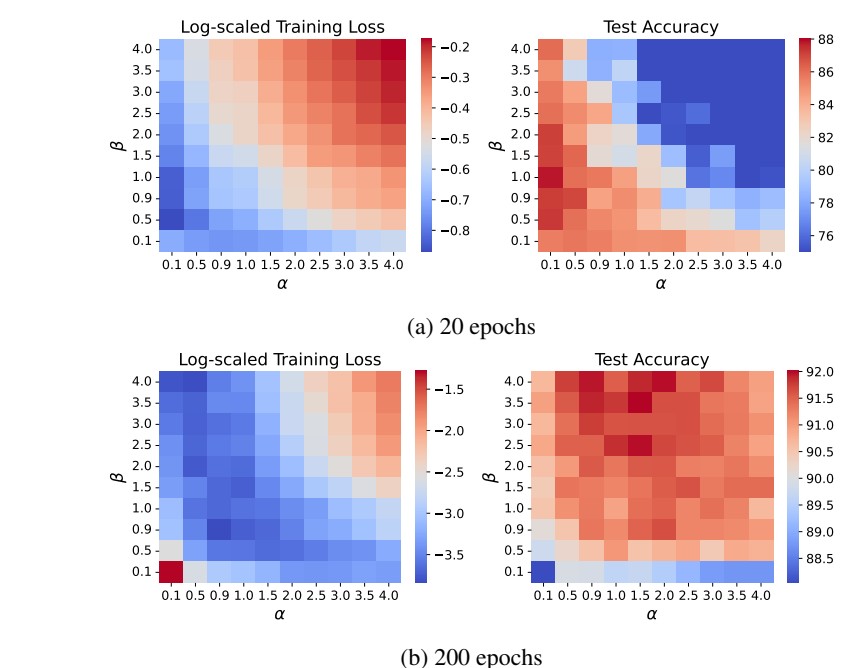

(a) 20 epochs

(b) 200 epochs

Figure 17: Log-scale training loss and test accuracies for $(\alpha, \beta)$ hyperparameters with ResNet18 on CIFAR10 at different epochs. Optimal learning rate $\gamma_0 = 10^{-3}$, weight decay $\lambda = 0.01$.

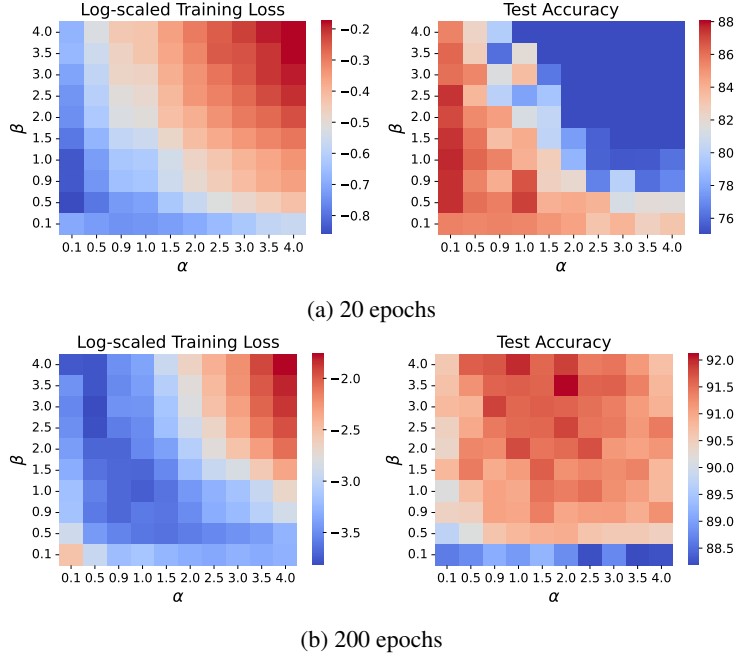

(a) 20 epochs

(b) 200 epochs

Figure 18: Log-scale training loss and test accuracies for $(\alpha, \beta)$ hyperparameters with ResNet18 on CIFAR10 at different epochs. Optimal learning rate $\gamma_0 = 10^{-3}$, weight decay $\lambda = 0$.

## F.5 COMPARISION WITH INNA

We evaluate INNA on GPT-2 Mini and compare it to INNAprop and AdamW. Following Castera et al. (2021), we used the recommended hyperparameters $(\alpha, \beta) = (0.5, 0.1)$ and tested learning rates

$\{1e-4, 1e-3, 1e-2, 1e-1\}$, selecting $\gamma_0 = 0.1$ as the best. Figure 19 shows that INNAprop and AdamW outperform INNA in both convergence speed and final validation loss.

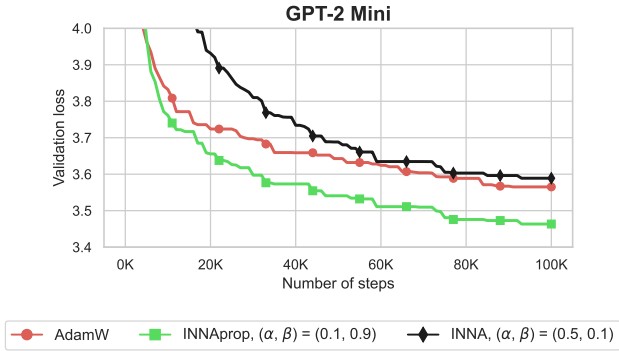

Figure 19: Validation loss comparison during GPT-2 mini training from scratch on the OpenWebText dataset.

# G EXPERIMENTAL SETUP

## G.1 CIFAR-10

We used custom training code based on the PyTorch tutorial code for this problem. Following standard data-augmentation practices, we applied random horizontal flips and random offset cropping down to 32x32, using reflection padding of 4 pixels. Input pixel data was normalized by centering around 0.5.

| Hyper-parameter | Value |
| --- | --- |
| Architecture | VGG11 and ResNet18 |
| Epochs | 200 |
| GPUs | 1×V100 |
| Batch size per GPU | 256 |
| Baseline LR | 0.001 |
| Seeds | 8 runs |

| Hyper-parameter | Value |
| --- | --- |
| Baseline schedule | cosine |
| Weight decay $\lambda$ | 0.01 |
| $\beta_1, \beta_2$ (for AdamW) | 0.9, 0.999 |
| $\sigma$ (for INNAprop) | 0.999 |

## G.2 FOOD101

We used the pre-trained models available on PyTorch for VGG11 and ResNet18.[5]

| Hyper-parameter | Value |
| --- | --- |
| Architecture | VGG11 and ResNet18 |
| Epochs | 200 |
| GPUs | 1×V100 |
| Batch size per GPU | 256 |
| Baseline LR | 0.001 |
| Seeds | 3 runs |

| Hyper-parameter | Value |
| --- | --- |
| Baseline schedule | cosine |
| Weight decay $\lambda$ | 0.01 |
| $\beta_1, \beta_2$ (for AdamW) | 0.9, 0.999 |
| $\sigma$ (for INNAprop) | 0.999 |

## G.3 IMAGENET

We used the same code-base as for our CIFAR-10 experiments, and applied the same preprocessing procedure. The data-augmentations consisted of PyTorch's RandomResizedCrop, cropping to 224x224 followed by random horizontal flips. Test images used a fixed resize to 256x256 followed by a center crop to 224x224.

---

[5] https://pytorch.org/vision/stable/models.html

### G.3.1  RESNET18

| Hyper-parameter | Value |
|---|---|
| Architecture | ResNet18 |
| Epochs | 90 |
| GPUs | 4×V100 |
| Batch size per GPU | 64 |
| Baseline LR | 0.001 |
| Seeds | 3 runs |

| Hyper-parameter | Value |
|---|---|
| Baseline schedule | cosine |
| Weight decay $\lambda$ | 0.01 |
| $\beta_1, \beta_2$ (for AdamW) | 0.9, 0.999 |
| $\sigma$ (for INNAprop) | 0.999 |

### G.3.2  RESNET50

| Hyper-parameter | Value |
|---|---|
| Architecture | ResNet18 |
| Epochs | 90 |
| GPUs | 4×V100 |
| Batch size per GPU | 64 |
| Baseline LR | 0.001 |
| Mixed precision | True |
| Seeds | 3 runs |

| Hyper-parameter | Value |
|---|---|
| Baseline schedule | cosine |
| Weight decay $\lambda$ | 0.1 |
| $\beta_1, \beta_2$ (for AdamW) | 0.9, 0.999 |
| $\sigma$ (for INNAprop) | 0.999 |

### G.3.3  VIT/B-32

| Hyper-parameter | Value |
|---|---|
| Architecture | ViT/B-32 |
| Epochs | 300 |
| GPUs | 8×A100 |
| Batch size per GPU | 128 |
| Baseline LR | 0.001 |
| Seeds | 5000 |

| Hyper-parameter | Value |
|---|---|
| Baseline schedule | cosine |
| Warmup | linear for 30 epochs |
| Weight decay $\lambda$ | 0.1 |
| $\beta_1, \beta_2$ (for AdamW) | 0.9, 0.999 |
| $\sigma$ (for INNAprop) | 0.999 |

### G.4  GPT2 FROM SCRATCH

We followed the NanoGPT codebase [6] and we refer to (Brown et al., 2020) as closely as possible, matching the default batch-size and schedule.

| Hyper-parameter | Value |
|---|---|
| Architecture | GPT-2 |
| Batch size per gpu | 12 |
| Max Iters | 100000 |
| GPUs | 4×A100 |
| Dropout | 0.0 |
| Baseline LR | refer to (Brown et al., 2020) |
| Warmup Steps | 500 |

| Hyper-parameter | Value |
|---|---|
| Seeds | 5000 |
| Weight decay $\lambda$ | 0.1 |
| $\beta_1, \beta_2$ (for AdamW) | 0.9, 0.95 |
| $\sigma$ (for INNAprop) | 0.99 |
| Gradient Clipping | 1.0 |
| Float16 | True |

### G.5  GPT-2 WITH LORA

We followed the LoRA codebase [7] and we refer to (Hu et al., 2021) as closely as possible, matching the default batch-size, training length, and schedule. We train all of our GPT-2 models using AdamW (Loshchilov & Hutter, 2017) and INNAprop on E2E dataset with a linear learning rate schedule for 5 epochs. We report the mean result over 3 random seeds; the result for each run is taken from the best epoch.

---

[6] https://github.com/karpathy/nanoGPT

[7] https://github.com/microsoft/LoRA

| Hyper-parameter | Value |
|---|---|
| Architecture | GPT-2 |
| Batch size per gpu | 8 |
| Epochs | 5 |
| GPUs | 1×A100 |
| Dropout | 0.1 |
| Baseline LR | 0.0002 |
| Warmup steps | 500 |

| Hyper-parameter | Value |
|---|---|
| Seeds | 3 runs |
| Weight decay $\lambda$ | 0.01 |
| $\beta_1, \beta_2$ (for AdamW) | 0.9, 0.98 |
| $\sigma$ (for INNAprop) | 0.98 |
| Learning Rate Schedule | Linear |
| LoRA $\alpha$ | 32 |

