# OpenReview forum: "A second-order-like optimizer with adaptive gradient scaling for deep learning"
_ICLR.cc/2025/Conference — Submitted to ICLR 2025_

### Official Review · Reviewer_BDCW · 2024-10-16

**Soundness:** 2
**Presentation:** 3
**Contribution:** 3
**Rating:** 6
**Confidence:** 3

**Summary:**

The paper combines RMSprop adaptive gradient scaling to the existing INNA optimizer, which is based on the Dynamical Inertial Newton. The obtained optimizer leverages second-order information while having similar computational requirements to first-order methods. With appropriate hyper-parameters the optimizer matches and in some cases outperforms AdamW.

**Strengths:**

The paper is well written, and easy to follow.
-    The derivation of the algorithm is well motivated.
-    They leverages second order information without requiring the hessian.
-    The algorithm seems to perform on par or even outperform AdamW in certain settings like language modeling.
-    The experiments cover a good amount of tasks between image classification and language modeling.
-    The authors open-source the code.

**Weaknesses:**

-   The introduction of the paper makes a big deal about the energy consumption of training large language models, but the authors don't measure or show how their method compares to the competitors in terms of wall-clock time in their experiments. Particularly interesting would be seeing how wall-clock time on ImageNet or language modeling tasks compare.
-   The method lacks comparison with more recent optimizers other than AdamW. Sophia for example is another fast and efficient second-order optimizer paper, which the author cite, but don't compare with. Lion is another recent and high performing optimizer, which was cited but not compared with either.
-   Numeric results are reported without standard deviation/standard error or confidence intervals

**Questions:**

-   How does INNAprop compare in terms of wall-clock time to AdamW or Sophia?
-   How does INNAprop compare to Sophia/Lion and other more recent high performing optimizers?
-   Is the increase in test accuracy of INNAprop compared to AdamW statistically significant? What are the error ranges?

---

> ### Author Response · Authors · 2024-11-24
> **Response to Reviewer BDCW**
>
> We thank the reviewer for his comments and constructive feedback. Comments are answered in detail below. Please let us know if you have any follow-up questions.
>
> ### Q.1 How does INNAprop compare in terms of wall-clock time to AdamW or Sophia?
>
> Thank you for the question and suggestion. We will definitely include additional experiments to directly compare the wall-clock time of INNAprop, AdamW, and Sophia.
>
> We would like to mention that when comparing AdamW and INNAprop, both algorithms require the same amount of memory, with INNAprop introducing only one additional subtraction operation, which we found to have a negligible effect on training time on our preliminary tests.
>
> Note as well that for Sophia, a direct comparison is more complex, as its runtime is heavily influenced by the frequency of Hessian updates, which is a tunable hyperparameter.
>
>
> **Average wall-clock time per step on GPT-2**
>
> | Model      | Optimizer                       | Wall-Clock Time (ms) |
> |------------|---------------------------------|-----------------------|
> | GPT-2 small| AdamW                           | 433                |
> |            | INNAprop | 444         |
> |            | Sophia-G                          | 449                 |
>
> We report the average time per step of the three optimizers (AdamW, INNAprop, Sophia). The overall wall-clock time overhead is less than
> 4% compared with AdamW as seen in the Table above.
>
>
> ### Q.2 How does INNAprop compare to Sophia/Lion and other more recent high-performing optimizers?
>
> Sophia is presented by its authors as an optimizer specifically designed for large language model pre-training. Reproducing Sophia is challenging (see [this GitHub issue](https://github.com/Liuhong99/Sophia/issues/46)), but we followed the instructions in the paper and repository to the best of our ability. We added experiments with GPT-2 Small and observed that INNAprop outperforms both Sophia and AdamW on this experiment.
> This is Figure 5 in the new draft, we only tested GPT2-small because the authors of Sophia did not provide hypertuning for GPT-2 mini, as for GPT-2 medium it appears to be unstable.
>
> As for Lion, we cannot provide relevant experiments, we provided a detailed answer in the general comments.
>
> INNAprop, by contrast, is not tailored to a specific task or batch size. We evaluated it across a variety of benchmarks, including image classification and language modeling, both with and without training from scratch. We will discuss these differences further in our paper and include additional experiments with Sophia on GPT-2.
>
> ### Q.3 Is the increase in test accuracy of INNAprop compared to AdamW statistically significant? What are the error ranges?
>
> Thank you for the question. We report error bars in all cases except for GPT-2 training from scratch, where we performed a single run due to the high computational cost (as also done in the Sophia paper). For ImageNet experiments, we follow the approach used in the Lion paper, reporting the mean over 3 runs. For CIFAR-10, we provide results averaged over 8 runs.
>
> We could of course do more, but we have applied usual protocol comparisons of some of the latest and popular papers.
>
> Please also refer to our general comment outlining additional tables detailing INNAprop’s performance.

---

> > ### Comment · Reviewer_BDCW · 2024-11-25
> >
> > My concerns have been partly addressed. I still think that more optimizers like Lion should be compared in the paper. I am also not sure how the comments regarding energy consumption are relevant as INNAprop then performs on par with other optimizers in wall clock time.
> >
> > However, I think that INNAprop is an interesting optimizer concept, thus I have increased my score.

---

> ### Author Response · Authors · 2024-11-25
>
> We would like to thank you for your consideration and re-evaluation of the score.
>
> ### Q.4  I still think that more optimizers like Lion should be compared in the paper.
>
> Thank you for the suggestion. We agree that including comparisons with more optimizers like Lion would strengthen the paper. We will conduct additional experiments with Lion  and include the results as soon as possible.
>
> ### Q.5 I am also not sure how the comments regarding energy consumption are relevant as INNAprop then performs on par with other optimizers in wall clock time.
>
> While the wall clock time with other optimizers may appear similar, INNAprop achieves AdamW’s best performance in significantly fewer steps, meaning models require less training time to reach comparable results.
>
> As highlighted in Figure 6, INNAprop demonstrates clear advantages. On GPT-2 Small and Medium, INNAprop consistently outperforms AdamW in both perplexity and efficiency. For GPT-2 Large, AdamW requires 25% more iterations to match INNAprop’s performance. Please see the table below for details:
>
> **Performance comparison for GPT-2 finetuning with LoRA on the E2E Dataset (mean over 3 runs, lower is better).**
>
> | Model    | AdamW best perplexity test | INNAprop best perplexity test | Steps to match AdamW |
> |------------------|-----------------------------------|---------------------------------------|-------------------------------|
> | **GPT-2 Small**  | 3.48                            |3.44                                | 19,000 (31% faster)               |
> | **GPT-2 Medium** | 3.20                            | 3.17                                | 20,000 (25% faster)               |
> | **GPT-2 Large**  | 3.09                            | 3.06                                | 20,000 (25% faster)               |
>
> **Performance comparison for GPT-2 train from scratch on OpenWebText.**
> | Model          | AdamW Best          | INNAprop Best      | Steps to Match AdamW            |
> |-----------------|---------------------|--------------------|----------------------------------|
> | GPT-2 Mini     | 3.57                | **3.47**           | 51,000 ($1.96 \times$ faster)   |
> | GPT-2 Small    | 3.03                | **2.98**           | 79,000 ($1.26 \times$ faster)   |
> | GPT-2 Medium   | 2.85                | **2.82**           | 83,000 ($1.2 \times$ faster)    |
>
> Please let us know if you have any further questions and we would be happy to address them.

---

> > ### Comment · Reviewer_BDCW · 2024-11-25
> >
> > Thank you for highlighting this, apologize for forgetting about Figure 6. I would add the steps to match AdamW in a separate figure as well as it can be insightful.

---

> ### Author Response · Authors · 2024-11-25
>
> Thank you for the suggestion. We appreciate your feedback and will include the steps to match AdamW in a separate figure in the revised version.
>
> We hope we have addressed the concerns of the reviewer.  We are available for any further questions.

---

### Official Review · Reviewer_xYEd · 2024-11-03

**Soundness:** 2
**Presentation:** 3
**Contribution:** 3
**Rating:** 5
**Confidence:** 3

**Summary:**

The authors introduce a second-order optimization method, INNAprop, which combines the INNA method with RMSprop, while not increasing the memory-footprint. The method is evaluated on a few image classification benchmarks, and GPT-2 with the OpenWebText dataset, and directly compared to AdamW with somewhat favorable results.

**Strengths:**

It is a comprehensive and well-written paper that does a good job of explaining the motivation for, and the math behind, the proposed method. I do appreciate the main thesis of the work, and the effort that went into it. I like the discussion of, and approach to, hyperparameter tuning, as well as the comprehensive Appendix.

**Weaknesses:**

My main objection is the empirical evaluation. Firstly, the reported improvements (on test sets) are generally very marginal. Secondly, the results are not pushed to SOTA-level --- even for the chosen architectures (that are pretty old and pretty small). Thirdly, there are too few experiments to convince me that the proposed method really works better than, or even as well as, AdamW. Since the method is not faster, or less computationally demanding, I find myself unconvinced of its overall usefulness.

**Questions:**

I noticed that you mentioned the lack of computational resources two times (lines 259 and 359). You should know, that I am very sympathetic to that challenge. Still, my suggestion is that you bite the apple and invest that extra time and computational cost --- to really make a convincing case for your method :-)

---

> ### Author Response · Authors · 2024-11-24
> **Response to Reviewer Reviewer xYEd**
>
> We thank the reviewer for the time spent on our manuscript. We provide a detailed answer below and propose some modifications of the manuscript following the reviewer's advice.
>
> ### Q.1 Firstly, the reported improvements (on test sets) are generally very marginal.
>
> Please see the general comment where we add tables on INNAprop performances.
>
> ### Q.2 Secondly, the results are not pushed to SOTA-level --- even for the chosen architectures (that are pretty old and pretty small).
>
> Thank you for this observation. Our choice of benchmarks and architectures was guided by standard practice in the literature on optimization methods. For instance, popular works like D-Adaptation, Prodigy, AdaFisher, MARS, and Sophia also evaluate their performance on similar tasks and architectures such as ResNets, ViT and GPT-2 small. These benchmarks provide a fair and consistent comparison across optimizers. This is why we did that way.
>
> Additionally, due to limited computational resources, we focused on commonly used architectures and datasets that are computationally accessible while still providing meaningful insights into the performance of our method. This approach ensures that our results are comparable with prior work and feasible within our compute budget.
>
> We agree that exploring larger and more recent architectures would be valuable, and we see this as an important direction for future work.

---

### Official Review · Reviewer_9GYe · 2024-11-04

**Soundness:** 4
**Presentation:** 3
**Contribution:** 2
**Rating:** 6
**Confidence:** 3

**Summary:**

This paper proposes a new first-order optimization algorithm: INNAprop. The algorithm combines the INNA method with RMSProp to derive a novel optimization scheme. The authors evaluate INNAprop empirically against AdamW, the de facto standard deep learning optimizer. They explore a set of diverse tasks spanning image classification and natural language processing using several different classes of deep learning architectures.

**Strengths:**

I thought the introduction was strong and successfully explained the need for improved optimization algorithm algorithms in terms of real-world impact. The authors back this up by deriving a memory-efficient version of their algorithm matching the memory overhead of existing approaches.

A thorough discussion of the algorithm and how it can be derived is provided. The authors also discuss comparisons between INNAprop and several other methods in the literature, in terms of its derivation. This is a valuable contribution that helps to place the proposed method among existing work. On the surface, I felt that the combination of the two methods could be of limited novelty. However, after reviewing the derivation I feel this constitutes a reasonably strong contribution.

The paper is clear and well-written. The authors provide a thorough set of diverse experiments to investigate the proposed method and include multiple runs to account for variability. From my review of the manuscript, all results look sensible and technically sound.

**Weaknesses:**

Claiming second-order without computing second-order statistics doesn't seem accurate to me. The finite difference approximation may not be sufficiently accurate to achieve these benefits. However, I am not overly familiar with this approximation that the authors adopt from prior work. One way to alleviate this concern would be to provide additional justification for this terminology. Or, to include empirical evidence that the finite-difference approach is providing a good approximation of the actual second order statistics.

One of the claimed contributions is that INNAprop "matches or outperforms AdamW in both training speed and final accuracy on benchmarks". However, Fig. 2 shows the faster training speed of AdamW (the final error bars overlap for test accuracy so I'd agree that they match performance). Fig. 3 shows INNAprop as faster to converge for ResNet, but slower once again for ViT. The test accuracy is once again matched in both experiments. Fig. 4 shows a similar convergence speed with a modest (~1%) increase in accuracy for INNAprop. Finally, Fig. 5 shows a modest win for INNAprop on the smaller settings and matched performance for the larger model. For the LoRA finetuning (Fig. 6), there is a small win for the small setting and INNAprop matches AdamW on the medium and large setting.

The improvement over Adam variants is small across the board. The notable wins are on vision finetuning (Fig. 4) and small-scale GPT-2 training (Fig. 5). There are many methods that propose similarly slim margins and none are adopted by the community. Furthermore, under scrutiny from third-party implementations, the gains are often difficult to replicate [1]. Sadly, my honest opinion is that this method will not gain traction with the community --- it is unclear when it should be used over AdamW.

While often spurious, it is common for optimization algorithms to include theoretical analysis of their convergence. I do not consider this a critical weakness of this paper.

[1] "Descending through a Crowded Valley — Benchmarking Deep Learning Optimizers", Schmidt et al.

**Questions:**

Can you explain how you "systematically favored AdamW through the choice of recommended hyperparameters (scheduler, learning rates, weight decay)"? In more established tasks, I feel this is a reasonable claim. However, where INNAprop showed wins (finetuning on an uncommon dataset and training tiny GPT-2 models), I expect that standard AdamW hyperparameters are suboptimal. How much compute budget was spent tuning AdamW compared to INNAprop? I am also concerned that two different settings of INNAprop are shown in all experiments against a single choice of hyperparameters for AdamW.

In the appendix, you derived the DINAdam variant. Do you also explore this algorithm empirically?

With my initial review, I am recommending that the paper be rejected. I believe the paper is technically correct and the proposed algorithm is sufficiently novel. However, I do not believe that the algorithm provides a meaningful practical contribution for the wider community at this time. My concern is that a practitioner would always choose AdamW over INNAprop due to the marginal expected gains, increased risk of failure, and the additional overhead of parameter tuning required. My score would change if the authors could provide clear guidance on when a practitioner would choose INNAprop over AdamW and expect a win.

Minor points:

- "we provide quite extensive experiments" vague statement in abstract. I would prefer that this be made more precise.
- "extra second-order geometrical intelligence" vague technical term. What do you mean by this? I think this should be clarified in the main text.

-------
**Post-rebuttal:**

Following the response from the authors, I have increased my score (and reduced my confidence). This is due primarily to the author's point that several results that I considered to be minimal improvements are in fact significant relative to the current state-of-the-art.

I retain concerns that the proposed algorithm is unlikely to see widespread community adoption. However, the paper is sound and proposes a novel technique. I would prefer that we present novel and correct methods to the community for their consideration.

---

> ### Author Response · Authors · 2024-11-24
> **Response to Reviewer 9GYe (1/3)**
>
> We thank the reviewer for his feedback and propose a detailed response to his comments below.
>
> ### Q.1Claiming second-order without computing second-order statistics doesn't seem accurate to me. The finite difference approximation may not be sufficiently accurate to achieve these benefits. However, I am not overly familiar with this approximation that the authors adopt from prior work.
>
> It can be proved that dynamics of the type:
>
> $$
> \epsilon x''(t) + \alpha x'(t) + \beta \nabla^2J(x(t))x'(t)  + \nabla J(x(t)) = 0
> $$
>
> generate curves $x_{\alpha,\epsilon}$ that converge (under appropriate assumptions) to a solution of
>
> $$
> \beta \nabla^2 J(x(t))x'(t) + \nabla J(x(t)) = 0,
> $$
> i.e,
> $$
> x'=-\beta\nabla^2 J(x)^{-1}\nabla J(x),
> $$
> which is a continuous-time version of Newton's method (see [Alvarez et al., 2002] for some elements).
>
> It is, therefore, not unreasonable to consider that finite differences of approximate methods resemble Newton's method or exhibit some form of second-order intelligence. We can present results in this vein in our revision.
>
> ### Q.2 One of the claimed contributions is that INNAprop "matches or outperforms AdamW in both training speed and final accuracy on benchmarks." However, Fig. 2 shows the faster training speed of AdamW (the final error bars overlap for test accuracy so I'd agree that they match performance). Fig. 3 shows INNAprop as faster to converge for ResNet, but slower once again for ViT. The test accuracy is once again matched in both experiments. Fig. 4 shows a similar convergence speed with a modest (~1%) increase in accuracy for INNAprop. Finally, Fig. 5 shows a modest win for INNAprop on the smaller settings and matched performance for the larger model. For the LoRA finetuning (Fig. 6), there is a small win for the small setting and INNAprop matches AdamW on the medium and large setting.
>
>
> Thanks a lot for this question our presentation was not clear enough.
>
> First, as a general response on test accuracies, we encourage the referee to refer to the tables in the general comments, which show that INNAprop consistently outperforms AdamW across CIFAR-10, Food-101, ImageNet, and language modeling experiments.
>
> We now answer precisely to the referee's questions below:
>
> Figure 2: We do not understand the comment, as on the right-most figure the best training accuracy is INNAprop $(\alpha=0.1, \beta=0.9)$.
>
> Figure 3: The referee is right for that figure; however, with a weight-decay equal to $\lambda=0.01$, we achieve faster test and training accuracy over roughly 250 epochs (see Figure 15 which was present in our original submission). We feel this is a fair comparison due to the fact that we use a SOTA tuning for AdamW.
>
> Figure 4 and 5: we kindly disagree that a ~1% increase in test accuracy is modest. For instance, the optimizer Lion (Chen et al., 2023) or the work by Defazio et al. in “The Road Less Scheduled” achieves similar or smaller gains, yet these are considered significant in the field. Accuracy improvements at this scale, often measured in tenths of a percent, are particularly meaningful for benchmark datasets and competitive tasks.
>
> Figure 6: we kindly disagree, there is a clean win on several aspects, on GP2-small, medium. As for GPT2-large, AdamW needs 25% more iterations to reach INNAProp values. See table below:
>
> **Performance comparison for GPT-2 finetuning with LoRA on the E2E Dataset (mean over 3 runs, lower is better)**
>
> | Model    | AdamW best perplexity test | INNAprop best perplexity test | Steps to match AdamW |
> |------------------|-----------------------------------|---------------------------------------|-------------------------------|
> | **GPT-2 Small**  | 3.48                            |**3.44**                                | 19,000 (31% faster)               |
> | **GPT-2 Medium** | 3.20                            | **3.17**                                | 20,000 (25% faster)               |
> | **GPT-2 Large**  | 3.09                            | **3.06**                                | 20,000 (25% faster)               |
>
> **Performance comparison for GPT-2 train from scratch on OpenWebText.**
> | Model          | AdamW Best          | INNAprop Best      | Steps to Match AdamW            |
> |-----------------|---------------------|--------------------|----------------------------------|
> | GPT-2 Mini     | 3.57                | **3.47**           | 51,000 ($1.96 \times$ faster)   |
> | GPT-2 Small    | 3.03                | **2.98**           | 79,000 ($1.26 \times$ faster)   |
> | GPT-2 Medium   | 2.85                | **2.82**           | 83,000 ($1.2 \times$ faster)    |

---

> ### Author Response · Authors · 2024-11-24
> **Response to Reviewer 9Gye (2/3)**
>
> ### Q.3 Can you explain how you "systematically favored AdamW through the choice of recommended hyperparameters (scheduler, learning rates, weight decay)"? In more established tasks, I feel this is a reasonable claim. However, where INNAprop showed wins (finetuning on an uncommon dataset and training tiny GPT-2 models), I expect that standard AdamW hyperparameters are suboptimal. How much compute budget was spent tuning AdamW compared to INNAprop? I am also concerned that two different settings of INNAprop are shown in all experiments against a single choice of hyperparameters for AdamW.
>
> We thank the reviewer for this question which is absolutely essential and that we have tried to better explain. In all our experiments, we either optimized AdamW’s hyperparameters through grid search or used the values recommended in recent literature, as explicitly stated when applicable.
> We provided a summary in Table 1 of the current draft, to explain this in a more direct way and we will improve this aspect in our revision.
>
> The only instance where we optimized INNAprop’s hyperparameters was for the friction parameters $(\alpha, \beta)$ pair in CIFAR-10 experiments (see Table 1 and Figure 1), fixing the other hyperparameters (learning rate, weight decay...) from SOTA tuning of AdamW.
>
> For larger tasks—such as ImageNet, GPT-2 on OpenWebText, and GPT-2 with LoRA, we directly applied hyperparameters recommended for AdamW in the literature and used the same settings for INNAprop.
>
> Given our limited computing resources, we aimed to present INNAprop's typical performance with "minimal" tuning, assuming that AdamW’s hyperparameters are near-optimal given their widespread use in recent studies (e.g., NanoGPT, Sophia, SchedulerFree, MARS).
>
> Of course we expect extensive tuning will further improve INNAprop's performance, especially for larger-scale experiments.
>
> ### Q.4 In the Appendix, you derived the DINAdam variant. Do you also explore this algorithm empirically?
>
> Yes, we explored the DINAdam variant numerically, as outlined in Appendix C.3. Incorporating RMSprop in a way similar to Adam, using momentum, results in an approach very close to NAdam (Dozat, 2016). Given this similarity and inferior behavior, we decided not to display the results in the main paper. We will add experiments on Appendix C.3. Note that DINAdam with $\beta=0$ and $\alpha=1$, we obtain exactly AdamW update rules.
>
> ### Q.5 My concern is that a practitioner would always choose AdamW over INNAprop due to the marginal expected gains, increased risk of failure, and the additional overhead of parameter tuning required. My score would change if the authors could provide clear guidance on when a practitioner would choose INNAprop over AdamW and expect a win.
>
> We agree that the recommendation for practitioners is not clear, thanks a lot for this question. We correct this in the revision.
>
> --- For fine-tuning (image classification and language modeling), INNAprop using AdamW hyperparameters gives better results than AdamW.
>
> --- For short duration, the choice $(\alpha=0.1, \beta=0.9)$, we expect a win as we see in the experiments.
>
> --- For long training, further hypertuning is necessary to provide trustworthy recommendations. At this stage, we think increasing $\alpha$ and $\beta$ will avoid overfitting as we see in CIFAR10 with $(\alpha=0.1, \beta=0.9)$. That's why we recommend to use $(\alpha=2.0, \beta=2.0)$ which is more slow to converge but give a better generalization than AdamW (see Figure 2).
>
> See also paragraph "Conclusion and recommendation for image classification" in Section 3.2.

---

> ### Author Response · Authors · 2024-11-24
> **Response to Reviewer 9GYe (3/3)**
>
> ### Q.6 "we provide quite extensive experiments" vague statement in abstract. I would prefer that this be made more precise.
>
> Thank you for your feedback. We address this in the revised version of the paper:
>
> "We evaluate INNAprop through extensive experiments on image classification tasks, using CIFAR-10, Food101, and ImageNet datasets with ResNets, VGG, DenseNet, and ViT architectures. For language modeling, we use GPT-2 on the OpenWebText dataset and fine-tune with LoRA on the E2E dataset."
>
> ### Q.7  "extra second-order geometrical intelligence" vague technical term. What do you mean by this? I think this should be clarified in the main text.
>
> Thank you for pointing these out. We shall address this further in the revision, we started addressing this question in our general comments on AdamW & INNAProp as well as the first answer to 9GYe. INNAProp may be indeed seen as a family of algorithms parameterized by $(\alpha, \beta)$, a dynamical system that has be proved to have many properties, including relationships with Newton's method [1] and fast method à la Nesterov [2,3]. This is why we used the term "second-order intelligence".
>
> [1] Felipe Alvarez, Hedy Attouch, Jérôme Bolte, and Patrick Redont. A second-order gradient-like dissipative dynamical system with hessian-driven damping.: Application to optimization and mechanics. Journal de mathématiques pures et appliquées, 81(8):747–779, 2002.
>
> [2] Hedy Attouch, Zaki Chbani, and Hassan Riahi. Rate of convergence of the nesterov accelerated gradient method in the subcritical case α ≤ 3. ESAIM: Control, Optimisation and Calculus of Variations, 25:2, 2019.
>
> [3] Hedy Attouch, Zaki Chbani, Jalal Fadili, and Hassan Riahi. First-order optimization algorithms via inertial systems with hessian driven damping. Mathematical Programming, pp. 1–43, 2022.

---

> ### Comment · Reviewer_9GYe · 2024-11-25
>
> Thank you for the detailed response.
>
> **Q1.** Understood. My concern was in assessing how unreasonable the finite difference approximation may be. But this is a minor point.
>
> **Q2.** Thank you for the clarifications.
>
> Regarding Fig 2, I was referring to convergence on the training loss. The optimizer is not solving for high train/test accuracy but low training loss. For this, Adam converges faster. To preempt, I understand that the loss is a surrogate for accuracy and we care about generalization performance too. But on many large-scale tasks training loss convergence remains the critical target.
>
> I think your points regarding other figures are fair. I am not overly familiar with the expected optimization trajectories for these tasks and it is possible I have assigned a lower value to the results than they are worth.
>
> **Q3.** Thank you for the detailed response. This is more or less what I expected and understood from the original version.
>
> **Q4.** Thanks for the clarification. Given that the algorithm is not exactly the same as NAdam, I believe that including results in Appendix C.3. is worthwhile.
>
> **Q5.** I remain hesitant to accept that practitioners would be ready to adopt INNAprop based on the empirical results and the introduction of two hyperparameters for which "tuning significantly impacts training". Especially as there is large variety in valid settings for these values across the different tasks explored. I believe that Adam(W) remains a dominant choice because for almost all modern deep-learning problems you can just grab the default settings and tweak the learning rate a little for competitive performance. And users are already familiar with this approach.
>
> Of course, the dominance of Adam should not rule out research into any alternatives. But I think the bar is naturally high after many years and many new optimization algorithms that remain untouched.
>
> **Q7.** This was a minor point that I wouldn't push to be addressed necessarily. My issue was more with "geometrical intelligence" than "second-order". This term is vague and suggesting intelligence of optimization algorithms is potentially misleading.
>
>
> Summarily, I appreciate the clarifications from the authors particularly around the empirical results. I am increasing my score (while decreasing my confidence) to reflect my potential miscalculation on the relative performance gained.

---

> ### Author Response · Authors · 2024-11-25
>
> We thank the reviewer for their thoughtful feedback and for taking the time to evaluate our paper. We also greatly appreciate your reconsideration of the score based on the clarifications provided.
>
> We will address all your points in the revised version, incorporating your valuable suggestions. Specifically:
> 1. We agree that including results for DINAdam in Appendix C.3 is worthwhile, given its distinct characteristics compared to NAdam, and we will add them to the updated version.
>
> 2.	We will further clarify terms like “geometrical intelligence” to ensure precision and avoid ambiguity.
>
> 3.	To address concerns regarding hyperparameter tuning, we provide clearer guidance on selecting $(\alpha, \beta)$ for INNAprop and discuss its impact in greater detail across different tasks.
>
> Please do not hesitate to reach out with any further questions or suggestions—we would be happy to address them.
>
> Sincerely,
>
> The Authors

---

### Official Review · Reviewer_Mr5G · 2024-11-05

**Soundness:** 2
**Presentation:** 2
**Contribution:** 2
**Rating:** 3
**Confidence:** 3

**Summary:**

This paper introduces INNAprop, a combination of the INNA optimizer with RMSprop gradient scaling. The resulting method has memory requirements similar to methods such as AdamW. Experiments show that this optimizer is competitive with AdamW.

**Strengths:**

The derivation of the algorithm is clear and the performance often seems competitive with AdamW. Large-scale models such as ViT-B/32 and GPT-2 were trained.

**Weaknesses:**

The INNAprop algorithm seems like a relatively straightforward extension from INNA. From what I can tell Algorithm 2 can be derived directly from the INNA equations in Table 2 and adding the factor $\\frac{1}{\\sqrt{v_{k+1}} + \\varepsilon}$ to the gradient. The motivation from a continuous dynamical systems perspective and the derivation done in appendix B seem equivalent to those from the INNA paper. This means that the theoretical insights from this paper seem limited.

This in itself wouldn't be a bad thing if the paper could make strong experimental claims about this small tweak to INNA leading to consistently stronger performance. However, this seems difficult to conclude for a few reasons, such as:

* No comparisons were made with INNA (without the RMSprop scaling) or other algorithms (Adam, ADAGRAD, SGD)
* In figures 3 and 6 INNAprop is sometimes indistinguishable from AdamW

Another surprise is that the optimal values for $\\alpha$ and $\\beta$ seem quite different to those found in the INNA paper (which ends up using $(0.1, 0.1)$ and $(0.5, 0.1)$) and it would be great to understand why. (In general, I am not convinced that the comparison with AdamW was the way to go, given that the proposed algorithm is far more similar to INNA. It would be insightful to know how the RMSprop extension changes INNA's behaviour and the choice of optimal hyperparameters for INNA.)

I understand that with limited compute it can be difficult to run large amounts of experiments, but I don't think that in it's current form this paper really provides new insights to the community: The RMSprop addition to INNA is pretty straightforward, and whether it really makes a difference is still hard to say.

**Questions:**

See above.

---

> ### Author Response · Authors · 2024-11-24
> **Response to Reviewer Mr5G (1)**
>
> We thank the reviewer for his work, we provide a detailed answer to his remarks below.
>
> ### Q.1 The INNAprop algorithm seems like a relatively straightforward extension from INNA.
>
> We believe this is a matter of perspective. INNAprop builds upon INNA in a manner similar to how past optimizers have evolved incrementally. For example, gradient descent (19th century) led to the heavy ball method (Polyak, 1964), which subsequently led to Nesterov acceleration (Nesterov, 1983), which then led to FISTA (2009). Likewise, Shor’s method (1985) inspired AdaGrad (2011), which then evolved into RMSprop (2012) with very minor changes and to Adam (2014), which is just HBF with RMSProp.
>
>
> ### Q.2 No comparisons were made with INNA (without the RMSprop scaling) or other algorithms (Adam, ADAGRAD, SGD).
>
> We agree with the reviewer. Although we conducted the experiments before starting the submission of the article, we did not report them. Specifically, we compared INNA on CIFAR-10 and GPT-2 small and found that it performed significantly worse than both INNAprop and AdamW -- including on time-clock. For this reason, we excluded INNA from our benchmarks.
> We will add the relevant experiments to the paper to illustrate this.
>
> ### Q.3 In Figures 3 and 6, INNAprop is sometimes indistinguishable from AdamW.
>
> Please see the general comment where we add tables on INNAprop performances in the revised version. They show more clearly the efficiency of our method.
>
> Figure 3: We agree with the reviewer. To clarify performance differences, we add a summary table in the revised paper showing each optimizer’s performance (see Table 2 and Table 3).
>
> Figure 6: The final performances are indeed close.
>
> But INNAprop ($\alpha=0.1,\beta=0.9)$ reaches AdamW’s peak performance much earlier in training. INNAprop ($\alpha=2.0,\beta=2.0)$ achieves better test accuracy performance than AdamW. See Table 2 in the revised version.
>
> --- on GPT-2 small, medium and large, INNAprop achieves a 25% reduction in training time, this will be discussed in the paper.
>
> For an illustration, refer to the table in the general comments.
>
> ### Q.4 Another surprise is that the optimal values for $\alpha$ and $\beta$ seem quite different from those found in the INNA paper.
>
> There are two factors that can explain  this difference
>
> --- Implementation.  Excluding the RMSprop term, our implementation of INNA differs from the original [1] by saving an additional memory slot.
>
> --- Gradient proxy. The other difference is of course that we use RMSProp as a gradient proxy which has a complex geometrical impact [2].
>
> This adjustment means that the optimal values of $(\alpha, \beta)$ for INNA and INNAprop are not directly comparable.
>
> Due to the time constraints of the rebuttal period, we are unable to conduct additional runs on INNA at this moment. However, we will perform multiple runs for different benchmarks, including INNA, as per your suggestion and include the results in the updated version of the manuscript.
>
> [1] Camille Castera, Jérôme Bolte, Cédric Févotte, and Edouard Pauwels. An inertial newton algorithm for deep learning. The Journal of Machine Learning Research, 22(1):5977–6007, 2021
>
> [2] Tieleman, T., & Hinton, G. (2012). Lecture 6.5 - rmsprop: Divide the gradient by a running average of its recent magnitude. Coursera: Neural Networks for Machine Learning.

---

> ### Author Response · Authors · 2024-11-25
> **Response to Reviewer Mr5G (2)**
>
> Dear Reviewer Mr5G,
>
> We conducted experiments comparing INNA, INNAprop, and AdamW on GPT-2 mini, trained from scratch on the OpenWebText dataset. For INNA, we used the hyperparameters recommended by its authors, specifically $(\alpha, \beta) = (0.5, 0.1)$, as stated in [1]:
>
> “Thus, INNA looks quite stable with respect to these hyperparameters. Setting $(\alpha, \beta) = (0.5, 0.1)$ appears to be a good default choice when necessary.”
>
> To ensure a fair comparison, we also tested multiple learning rates $(\{1e-4, 1e-3, 1e-2, 1e-1\})$ for INNA and identified $\gamma_0 = 0.1$ as the most effective setting. Our findings show that INNA converges more slowly than both INNAprop and AdamW, with INNA achieving a worse validation loss. Detailed results for GPT-2 mini are provided in Figure 19 (in Appendix F.5).
>
> We observed similar behavior on image classification tasks, consistent with the findings in the original INNA paper (see Figure 3 in [1]), where INNA does not outperform Adam on CIFAR-10 test accuracies. We will include additional image classification results in a future update.
>
> We appreciate your feedback and remain available to address any further questions or suggestions you may have.
>
> Sincerely,
>
> The Authors
>
> [1] Camille Castera, Jérôme Bolte, Cédric Févotte, and Edouard Pauwels. An inertial Newton algorithm for deep learning. The Journal of Machine Learning Research, 22(1):5977–6007, 2021

---

### Author Response · Authors · 2024-11-24
**General comment to all reviewers (1/2)**

We thank all the reviewers for their constructive feedback, which will helped us a lot in improving our paper.

We agree that we did not properly emphasize several important points, starting with the hypertuning protocol and several performance reports which are essential to the understanding of this paper.

We, of course, hope the referees will take into account the new insights we bring to our work through their evaluations.

**AdamW is favored**.
AdamW has been systematically advantaged by using state-of-the-art parameters. These parameters are reused as is for INNAprop, and *only afterwards*, $\alpha$ and $\beta$ are optimized on CIFAR-10 (ResNet18, VGG11).

*We did not optimize these parameters for any of the other benchmark we adress* (we only observed that lower weigh-decay for ViT-B/32 seems better for INNAprop).

This is summarized below:

--Hyperparameter tuning strategy--

| **Parameters**              | **AdamW tuning**                | **INNAprop tuning**        | **Comparative advantage** |
|----------------------------|----------------------------------|----------------------------|----------------------------|
| Learning rate              | Literature or grid search tuning | Reused from AdamW          | AdamW favored             |
| Step size scheduler        | Literature                      | Reused from AdamW          | N/A                        |
| Weight decay               | Literature or grid search tuning | Reused from AdamW          | AdamW favored             |
| RMSprop parameter          | Default or literature            | Reused from AdamW          | AdamW favored             |
| Inertial parameters ($\alpha, \beta$) | N/A                            | Tuned on CIFAR-10          | N/A                        |

Despite this obvious handicap, INNAprop performs much better, especially considering the extensive tuning previously invested in AdamW.


In our coming revision, we will optimize INNAprop with tuning learning rates, weight decay and RMSProp parameters for CIFAR-10 experiments. Due to the long runtime of these experiments, we will add the results as soon as they are completed.


**AdamW & INNAprop**.
INNAprop is closely tied to Adam, as it incorporates an additional friction term that implicitly leverages the Hessian of RMSProp, as shown in Equation 5. Furthermore, by setting $\alpha=\beta = 1$, we empirically recover the behavior of AdamW. Experiments demonstrate that this tuning consistently aligns with AdamW, see Figure 7, 8, 9, suggesting that AdamW can be seen as a special case within the broader INNAprop family. This connection should have been emphasized in our pedagogical introduction. We have revised Remark 3 and Appendix B.1 to provide a clearer explanation. We have added illustrations in Figures 7, 8, and 9 in Appendix B.1.

---

> ### Author Response · Authors · 2024-11-24
> **General comment to all reviewers (2/2)**
>
> **More comparisons**. We have conducted or will conduct more tests with competitive optimizers such as Sophia or INNA. As for Lion, it does not align with our framework: for images, we are unable to scale to large mini-batches. Moreover, Lion is not well-suited for LLMs at this time (according to the authors).
>
>
> **Train from Scratch on ImageNet (mean over 3 runs)**
>
> | Model      | Optimizer                       | Top-1 Accuracy |
> |------------|---------------------------------|----------------|
> | ResNet18   | AdamW                           | 69.34          |
> |            | INNAprop $\alpha=0.1$, $\beta=0.9$ | **70.12**     |
> |------------|---------------------------------|----------------|
> | ResNet50   | AdamW                           | 76.33          |
> |            | INNAprop $\alpha=1.0$, $\beta=1.0$ | **76.43**     |
> |------------|---------------------------------|----------------|
> | ViT-B/32   | AdamW                           | 75.02          |
> |            | INNAprop $\alpha=0.1$, $\beta=0.9$ | **75.23**     |
>
> **Train from Scratch on CIFAR-10 (mean over 8 runs)**
>
> | Model       | Optimizer                      | Top-1 Accuracy |
> |-------------|--------------------------------|----------------|
> | ResNet18    | AdamW                          | 91.14          |
> |             | INNAprop $\alpha=2.0$, $\beta=2.0$ | **91.58**    |
> |-------------|--------------------------------|----------------|
> | VGG11       | AdamW                          | 90.79          |
> |             | INNAprop $\alpha=2.0$, $\beta=2.0$ | **90.99**    |
> |-------------|--------------------------------|----------------|
> | DenseNet121 | AdamW                          | 86.19          |
> |             | INNAprop $\alpha=0.1$, $\beta=0.9$ | **86.91**    |
>
> **Performance comparison for GPT-2 finetuning with LoRA on the E2E Dataset (perplexity test, mean over 3 runs)**
>
> | Model    | AdamW Best | INNAprop Best| Steps to match AdamW |
> |------------------|-----------------------------------|---------------------------------------|-------------------------------|
> | **GPT-2 Small**  | 3.48                            |**3.44**                                | 19,000 (31% faster)               |
> | **GPT-2 Medium** | 3.20                            | **3.17**                                | 20,000 (25% faster)               |
> | **GPT-2 Large**  | 3.09                            | **3.06**                                | 20,000 (25% faster)               |
>
> **Performance comparison for GPT-2 train from scratch on OpenWebText  (validation loss)**
> | Model          | AdamW Best          | INNAprop Best      | Steps to Match AdamW            |
> |-----------------|---------------------|--------------------|----------------------------------|
> | GPT-2 Mini     | 3.57                | **3.47**           | 51,000 ($1.96 \times$ faster)   |
> | GPT-2 Small    | 3.03                | **2.98**           | 79,000 ($1.26 \times$ faster)   |
> | GPT-2 Medium   | 2.85                | **2.82**           | 83,000 ($1.2 \times$ faster)    |
>
> If you have any further questions, please feel free to contact us. We are happy to provide additional clarification.

---

### Author Response · Authors · 2024-11-26
**General comment to all reviewers (3)**

Dear Reviewers,

We have uploaded a new version of the paper, incorporating your feedback and addressing the points raised in your reviews.

We have added figures to highlight the gains of INNAprop, as shown in Tables 2, 3, and 4. Experiments requiring more time, as mentioned in our responses, will be added once completed.

As the update deadline is tomorrow, we are available to make any further adjustments or address additional concerns. Please feel free to reach out with any suggestions.

---

### Meta-Review · Area_Chair_FULj · 2024-12-23

**Metareview:**

This paper studies a new optimization algorithm that combines the INNA method with the RMSprop adaptive gradient scaling. The resulting method has memory and computational requirements similar to AdamW.  The paper demonstrates that this algorithm performs somewhat better than AdamW algorithm.

The reviews for the paper are slightly negative. The main concern of the reviewers is limited novelty and weak empirical analysis. I agree with the reviewers that the novelty is somewhat limited. The authors need to at least provide more solid empirical analysis to justify the claim of outperforming AdamW.  The presentation of the paper is poor and can also be improved. I also think it is worth comparing the performance of algorithm with approaches like Shampoo (even though these approaches are expensive, it is worth comparing the quality-computation tradeoff). For these reasons, I recommend rejection.

**Additional Comments On Reviewer Discussion:**

The rebuttal failed to convince all the reviewers about the novelty of the method and empirical analysis. While a couple of reviewers increased the score, it was still failed to convince the rest. I think these are valid concerns and need to be addressed carefully by the authors.

---

### Decision · Program_Chairs · 2025-01-22

Reject